# Disynaptic cerebrocerebellar pathways originating from multiple functionally distinct cortical areas

Julia U Henschke[1,2], Janelle MP Pakan[1,2,3]*

[1]Institute of Cognitive Neurology and Dementia Research, Otto-von-Guericke-University, Magdeburg, Germany; [2]German Centre for Neurodegenerative Diseases, Magdeburg, Germany; [3]Center for Behavioral Brain Sciences, Universitätsplatz, Magdeburg, Germany

**Abstract** The cerebral cortex and cerebellum both play important roles in sensorimotor processing, however, precise connections between these major brain structures remain elusive. Using anterograde mono-trans-synaptic tracing, we elucidate cerebrocerebellar pathways originating from primary motor, sensory, and association cortex. We confirm a highly organized topography of corticopontine projections in mice; however, we found no corticopontine projections originating from primary auditory cortex and detail several potential extra-pontine cerebrocerebellar pathways. The cerebellar hemispheres were the major target of resulting disynaptic mossy fiber terminals, but we also found at least sparse cerebrocerebellar projections to every lobule of the cerebellum. Notably, projections originating from association cortex resulted in less laterality than primary sensory/motor cortices. Within molecularly defined cerebellar modules we found spatial overlap of mossy fiber terminals, originating from functionally distinct cortical areas, within crus I, paraflocculus, and vermal regions IV/V and VI - highlighting these regions as potential hubs for multimodal cortical influence.

## Introduction

The environment provides constantly updating sensory signals that must be acted upon for an animal to perform basic behaviors necessary for survival, including navigating through the environment, feeding/foraging, and other goal-directed behaviors. These perception and action loops require various sensory modalities (i.e. somatosensation, audition, and vision) to be integrated and translated into directed motor output. Both the cerebral cortex (*Stein and Stanford, 2008*) and the cerebellum (*Snider and Stowell, 1944*; *Rondi-Reig et al., 2014*; *Baumann et al., 2015*) are major brain regions involved in this sensorimotor integration and translation. Connections between these two brain regions form one of the largest projection pathways in the brain and selective expansion of this cortico-cerebellar system occurs across evolution (*Gutiérrez-Ibáñez et al., 2018*; *Smaers and Vanier, 2019*). This reflects the importance of cerebrocerebellar communication, however, the precise functions of these pathways are not fully understood (*Apps and Watson, 2013*). A vital initial step in understanding the role of cerebrocerebellar communication is to have a comprehensive map of the pathways linking these two structures as well as the precise organization of the termination of these pathways within the cerebellum.

Due to the indirect nature of cerebrocerebellar connections, it has been difficult to study their organization in precise detail. While a large body of research has mapped out the corticopontine projections (*Brodal and Bjaalie, 1997*; *Glickstein, 1997*; *Leergaard and Bjaalie, 2007*; *Proville et al., 2014*) or the pontocerebellar projections (*Pijpers and Ruigrok, 2006*; *Pakan et al., 2010*; *Proville et al., 2014*; *Biswas et al., 2019*), few studies have utilized neurotropic viruses

*For correspondence:
janelle.pakan@med.ovgu.de

Competing interests: The authors declare that no competing interests exist.

(*Kelly and Strick, 2003*; *Suzuki et al., 2012*). Therefore, the precise routes for information flow from the cortex to the cerebellum, including detailed terminal organization in the highly modular cerebellar cortex, have remained largely inferred. The integration of multimodal inputs to the cerebellum is a fundamental operation that would allow for the precise coordination of sensory-driven movements within this brain region. Anatomically, the potential for a single granule cell to receive multimodal input *via* both descending motor cortex and ascending proprioceptive pathways has been shown in mice (*Huang et al., 2013*). However, the potential for co-innervation originating from various cortical areas spanning multiple modalities is unknown, even on the regional level in the cerebellum, and has consequences for the role of cerebro-cerebellar-cerebro feedback loops in learning and predictive motor control (e.g. *Chabrol et al., 2019*).

Using a mono-trans-synaptic anterograde viral tracer (*Zingg et al., 2017*; *Zingg et al., 2020*), we investigated the precise cerebrocerebellar pathways from key sensory, motor, and association regions of the cortex *via* the pontine and other intermediate precerebellar nuclei to all regions of the cerebellar cortex (*Figure 1A–C*); ultimately providing a map of the potential pathways linking various functionally specific cortical regions with the cerebellum. Following injections into the primary motor (M1), somatosensory (S1), visual (V1), auditory (A1), posterior parietal association cortex (PPC), and the dorsal field of auditory cortex (AuD), we found a highly organized topography of labeled pontine cells, with the notable exception that injections into A1 produced only terminal labeling in the pons; indicating the lack of a A1-ponto-cerebellar pathway in mice. We quantified the number of resulting mossy fiber terminals and described their relationship to the internal organization of the cerebellum. The majority of labeled mossy fiber terminals from the primary sensory and motor cortical regions were in the contralateral cerebellar hemisphere, whereas from association cortices this laterality was less evident. Cortical influences were not restricted to the cerebellar hemispheres, as terminals spanned all regions of the cerebellar cortex, with biases depending on the cortical modality. Cerebellar subdivisions with the highest regional co-innervation of multimodal inputs were crus I, the paraflocculus (PFl), vermal lobule VI and lobules IV/V, highlighting the potential for modular multimodal processing of information originating from the cerebral cortex.

## Results

To examine the topography of cerebrocerebellar pathways from primary sensory and motor cortical regions as well as sensory association areas, we utilized an AAV1.cre construct that has been shown to act as a trans-synaptic anterograde tracer that crosses a single functional synapse (mono-trans-synaptic; *Zingg et al., 2017*; *Zingg et al., 2020*). We injected AAV1.cre into various cortical regions in tdTomato reporter mice and quantified the resulting labeled precerebellar pathways and mossy fiber terminals in the cerebellum (*Figure 1*). Target cortical areas included M1 (forelimb/hindlimb regions), S1 (forelimb/hindlimb regions), PPC, V1, A1, and AuD (*Figure 1A,D*; three mice per target region; see *Supplementary file 1A*).

### Anterograde tracing of indirect cerebrocerebellar pathways

Following cortical injections, mono-trans-synaptically labeled cells were found in the ipsilateral basal pontine nuclei (referred to as pontine nuclei throughout) from all cortical target regions, with the exception of A1 (*Figure 1E*). In contrast, after injections in the more dorsal secondary auditory region, AuD, a corticopontine pathway was observed (*Figure 1E*). The lack of corticopontine projections from A1, was confirmed by injecting CAV.cre into the pontine nuclei; this viral vector is preferentially taken up by axon terminals, resulting in retrograde labeling. Here we found retrogradely labeled cells in AuD and more ventral auditory cortex (AuV), but no labeled cells in A1 (*Figure 1—figure supplement 1*). Although anterogradely labeled cell bodies were not observed in the pontine nuclei following AAV1.cre injections in A1, labeled fibers were found in the dorsomedial portion of the ipsilateral pons (*Figure 1E*; *Supplementary file 1B*). These fibers could be either traversing through the pons, potentially towards lower brainstem structures (e.g., cochlear and vestibular nuclei see *Figure 2A*; see also *Budinger et al., 2000*), or disynaptic terminals resulting from labeled indirect pathways from A1 to the pontine nuclei, likely *via* the inferior colliculus (*Schuller et al., 1991b*; *Caicedo and Herbert, 1993*). Indeed, anterogradely labeled neurons were consistently observed in the inferior colliculus after A1 injections (*Figure 2B*) and following injection of AAV1.cre in the inferior colliculus, anterogradely labeled cells were found in the pontine nuclei as well as mossy fiber

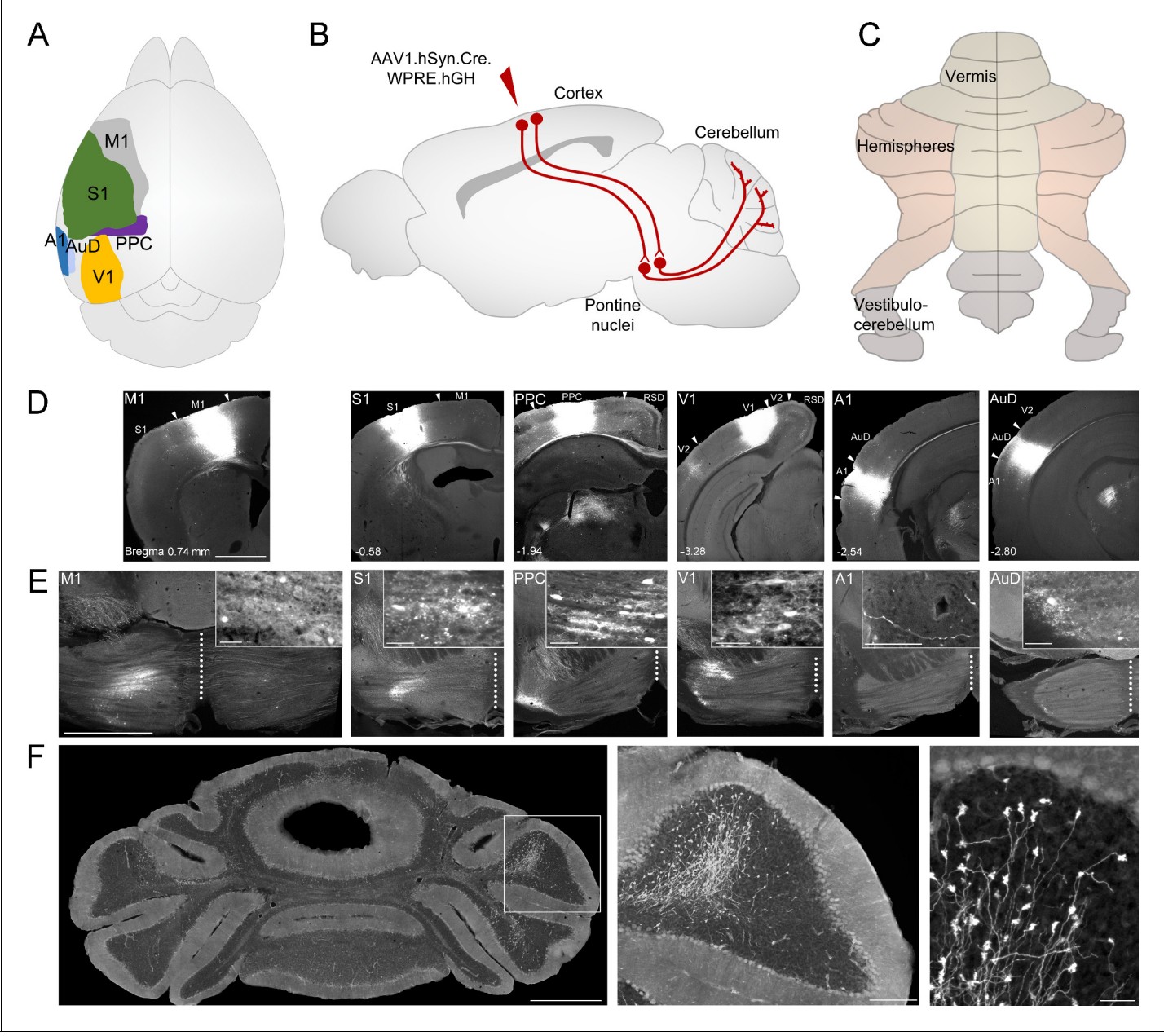

**Figure 1.** Anterograde tracing of indirect cerebrocerebellar pathways using a mono-trans-synaptic adeno-associated virus (AAV). (**A**) Schematic outlining cortical target areas for mono-trans-synaptic anterograde tracer injections: primary motor (M1), primary somatosensory (S1), posterior parietal association cortex (PPC), primary visual (V1), primary auditory (A1), and dorsal auditory (AuD) cortex. (**B**) Principle of mono-trans-synaptic anterograde tracing (e.g cortico-pontine-cerebellar pathway) using a specific adeno-associated virus (AAV1.cre). (**C**) Schematic of gross anatomical divisions of the unfolded mouse cerebellum (according to *Marani and Voogd, 1979*). (**D**) Images of coronal sections showing representative injection sites into M1, S1, PPC, V1, A1, and AuD (from left to right). Arrowheads indicate regional borders. Distance from bregma is indicated based on the mouse stereotaxic atlas (*Franklin and Paxinos, 2007*). (**E**) Images of coronal sections illustrating the mono-trans-synaptic labeling in the pontine nuclei (i.e. corticopontine fibers, postsynaptic pontine cells, pontine fibers) following injections into M1, S1, PPC, V1, A1, and AuD. Note that the medial-lateral topography of pontine labeling correlates with the rostral-caudal localization of these cortical regions. Injections into A1 resulted in labeled fibers within the pons, but no labeled cells. (**F**) Images of coronal section showing representative mossy fiber labeling following an injection into M1 at different magnifications. Scale bars 1 mm (D, E, F left), 20 μm (insets in E), 200 μm (F middle), 50 μm (F right). Retrosplenial Cortex (RSD); secondary visual cortex (V2). The online version of this article includes the following figure supplement(s) for figure 1:

**Figure supplement 1.** Retrogradely labelled cells following injection in pontine nuclei.

**Figure supplement 2.** Resulting anterograde labeling in the pontine nuclei and mossy fiber terminals following injection of mono-trans-synaptic adeno-associated virus (AAV) into the inferior colliculus.

*Figure 1 continued on next page*

*Figure 1 continued*

**Figure supplement 3.** Injection site quantification method and relationship between the injection site volume and the resulting mossy fiber labeling in the cerebellar cortex.

terminals in the cerebellum, demonstrating the potential for a trisynaptic A1-collicular-cerebellar pathway via the pontine nuclei (*Figure 1—figure supplement 2*).

In agreement with previous studies (*Leergaard and Bjaalie, 2007*; for review see, *Kratochwil et al., 2017*), resulting pontine labeling after cortical injections in mice was topographically organized with more rostral cortical regions projecting more medially in the pons and caudal cortical areas projecting towards the lateral extent of the pontine nuclei (*Figure 1E*, see also *Figure 3B*). While most pontine labeling was strictly ipsilateral, M1 was the only cortical target region that resulted in bilateral pontine labeling (average of 118 ± 36 neurons ipsilateral to 8 ± 3 neurons contralateral; *Figure 1E*). Previous anterograde tracer studies have observed corticopontine fibers largely in the ipsilateral pontine nuclei with relatively sparse labeling in the contralateral pons (*Mihailoff et al., 1985*; *Leergaard and Bjaalie, 2007*). However, it was unclear if these sparse contralateral projections were fibers of passage or axon terminals; our results suggest the later for M1 projections and the former for all other cortical targets regions described here. Interestingly, the existence of functional synapses for contralateral corticopontine projections was also reported in primates, however, also exclusively after M1 injections (*Morecraft et al., 2018*).

We observed anterogradely labeled cells in various other precerebellar nuclei (*Figure 2A*; *Supplementary file 1B*; see also *Ruigrok et al., 2015*). In fact, after S1 injections we found more labeled cells in extra-pontine precerebellar nuclei than in pontine nuclei (70 ± 8% of total labeling) and after M1 injections, just under half (40 ± 3%). This labeling spanned several precerebellar nuclei, including the red nucleus and reticulotegmental nucleus, as well as more caudal brainstem regions including the spinal trigeminal nucleus, lateral reticular nucleus, the matrix region x and the vestibular nuclei (*Figure 2*, *Supplementary file 1B*). While the reticulotegmental nucleus is often grouped together with the basal pontine nuclei, these regions in the pons appear to play different functional roles (*Cicirata et al., 2005*), hence, in this study we classify the reticulotegmental nucleus as a separate extra-pontine region. Additionally, we observed labeled fibers in both the medial and inferior cerebellar peduncles (see *Figure 2A*), indicating disynaptic tracts projecting to the cerebellum from pontine regions and more caudal extra-pontine precerebellar nuclei, respectively. Since there was no labeling observed in the pontine nuclei after A1 injections, the resulting mossy fiber terminals must reach the cerebellum through other precerebellar nuclei, of which we found labeling in the vestibular as well as the cochlear nuclei (see *Figure 2A*).

We did not observe labeled cells from any target cortical region in the spinal or lateral vestibular nuclei, the external cuneate nucleus, or the inferior olive. We did, however, observe fibers and terminal labeling in the inferior olive following injections in S1 (*Figure 2C*) and to a lesser extent after injections in M1. Interestingly, also only after S1 and M1 injections, anterogradely labeled cells were observed in the matrix region x, which has been suggested as a candidate preolivary relay nucleus (*Ackerley et al., 2006*). There is some controversy regarding the existence of direct cerebro-olivary connections in rodents, particularly originating from M1 (*Swenson et al., 1989*; *Baker et al., 2001*; *Ackerley et al., 2006*); for review see *Watson and Apps, 2019*); our results support disynaptic input from the cerebral cortex to inferior olivary neurons. Sparse fibers were also observed in the lateral cerebellar nucleus following S1 and M1 injections, likely from pontocerebellar collaterals (*Cicirata et al., 2005*; *Biswas et al., 2019*).

## Regional organization of cerebrocerebellar mossy fiber terminals

Cerebrocerebellar mossy fiber terminal labeling in the granule cell layer of the cerebellum was systematically present, regardless of the cortical origin (e.g. *Figure 1F*, *Supplementary file 1B*). We did not find AAV1.cre-induced tdTomato expressing granule cells or Purkinje cells in any of the cases, indicating that the AAV construct did not transfer to higher-order downstream structures beyond mono-synaptic connections (see also *Zingg et al., 2017*; *Zingg et al., 2020*). We quantified the number of resulting mossy fiber terminals throughout the cerebellum and found that this was highly variable across cortical regions, with S1 and M1 resulting in the largest number of labeled

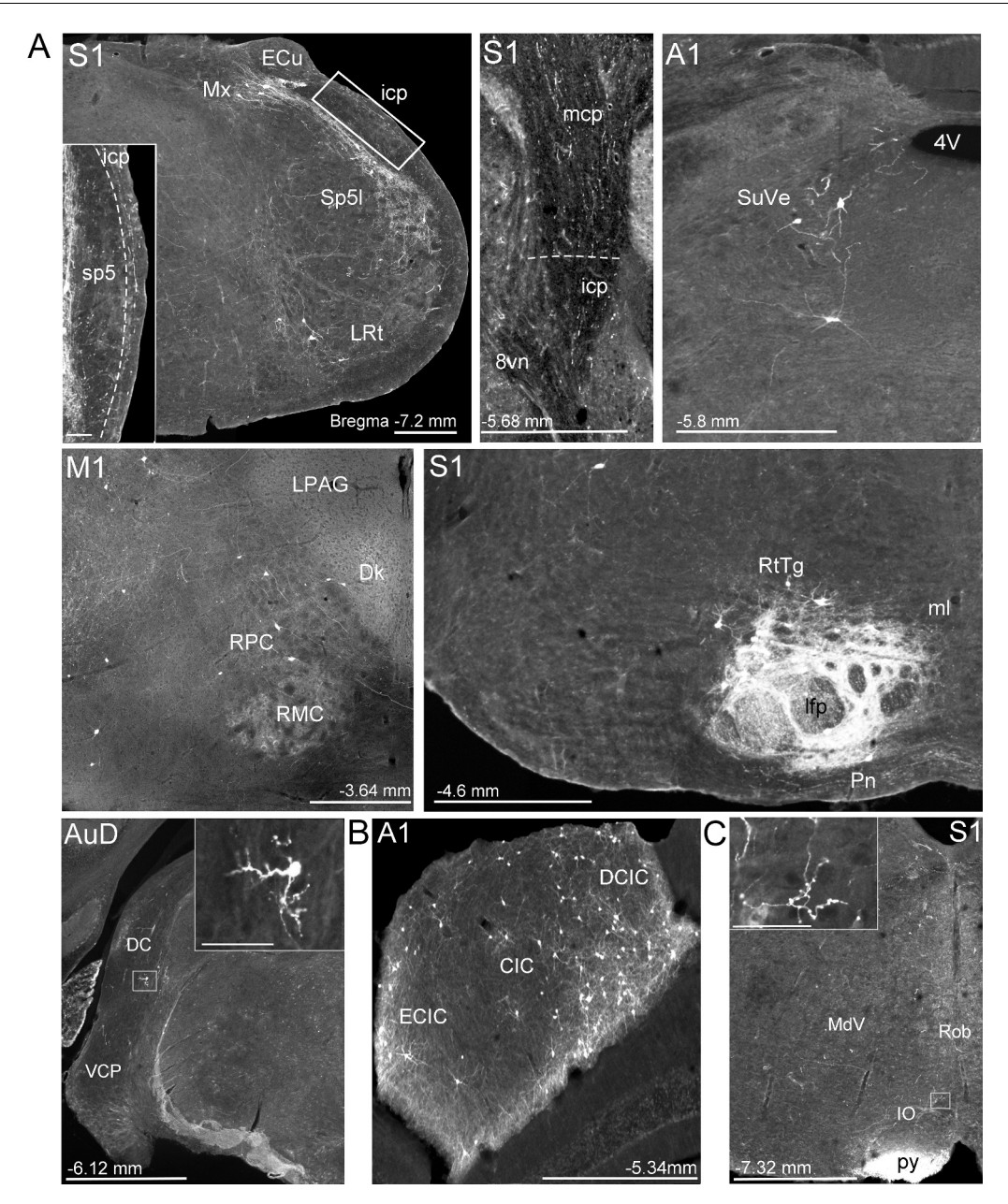

**Figure 2.** Extra-pontine labeling in key precerebellar nuclei following mono-trans-synaptic adeno-associated virus (AAV) injections in target cerebral cortical regions. (**A**) Representative labeling in precerebellar nuclei listed in *Supplementary file 1B*, after injections of AAV1.cre into primary motor (M1), primary somatosensory (S1), primary auditory (A1), and dorsal auditory (AuD) cortex. Images show labeled cells in interpolar part of the spinal trigeminal nucleus (Sp5I), lateral reticular nucleus (LRt), matrix region x (Mx), superior vestibular nucleus (SuVe), red nucleus (RPC, RMC), pontine reticulotegmental nucleus (RtTg), dorsal cochlear nucleus (DC) and ventral cochlear nucleus, posterior part (VCP), which provide alternative extra-pontine mossy fiber pathways for indirect cortical input to the cerebellum. Descending labeled fiber tracts can also be observed in the longitudinal fasciculus of the pons (lfp) and the pyramidal tracts (py). Further, fibers travelling to the cerebellar cortex in the middle and inferior cerebellar peduncles (mcp and icp, respectively) after S1 injections. (**B**) Labeled cells in the inferior colliculus (central [CIC], external cortex [ECIC] and dorsal cortex [DCIC] inferior colliculus) following A1 injections. (**C**) Labeled fibers in the ipsilateral inferior olive (IO) following S1 injections. For all, distance from bregma is indicated based on the mouse stereotaxic atlas (*Franklin and Paxinos, 2007*). Scale Bars 500 µm (**A**), 1 mm (**B, C**), 100 µm all insets. 4th ventricle (4V), vestibular nerve (8vn), nucleus of Darkschewitsch (Dk), lateral periaqueductal gray (LPAG), medial lemniscus (ml), medullary reticular nucleus, ventral (MdV), raphe obscurus nucleus (Rob).

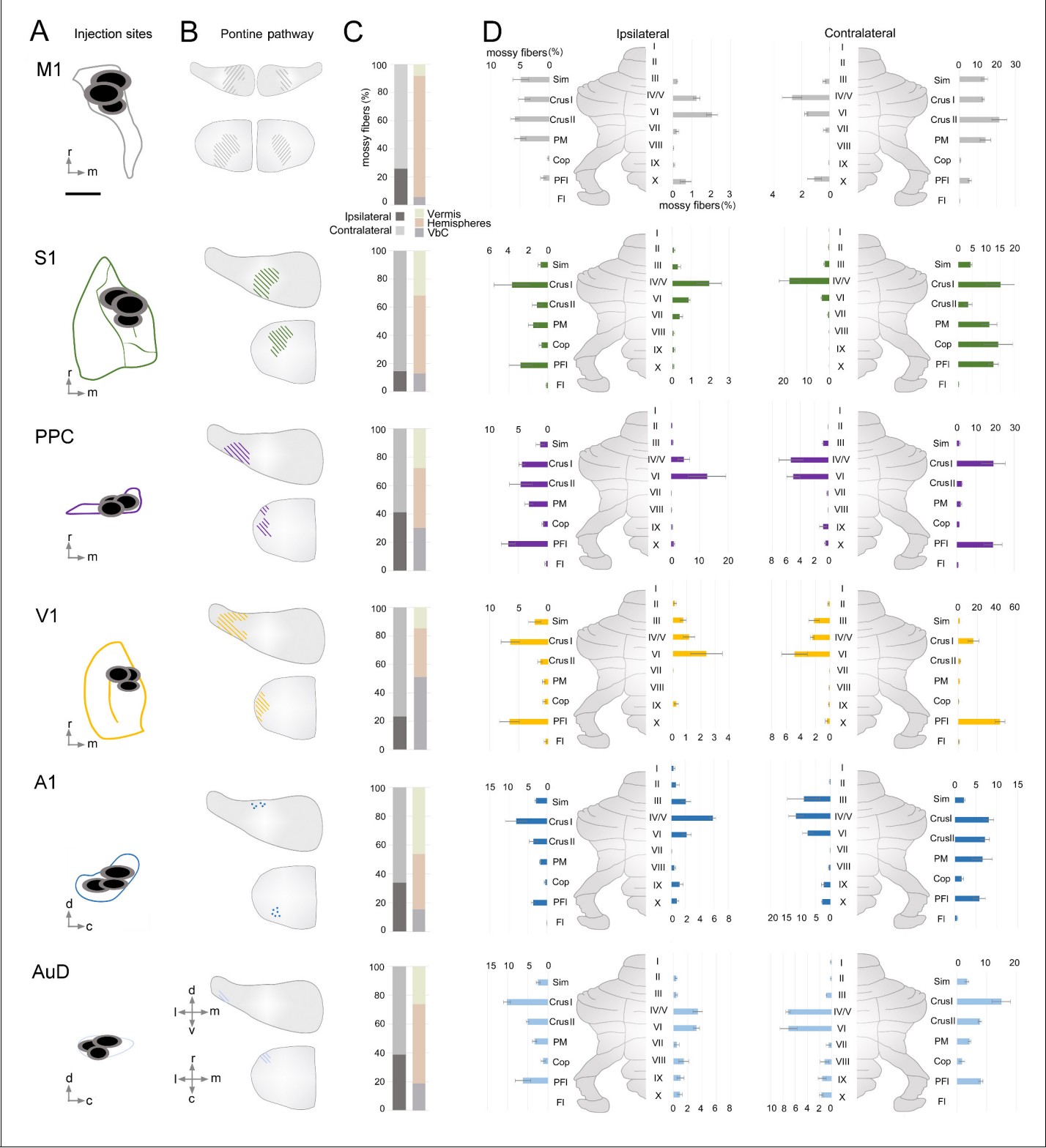

**Figure 3.** Organization of indirect cerebrocerebellar mossy fiber terminals in the cerebellum from target cerebral cortical regions. (**A**) Schematic of cortical target areas as outlined on the cortical surface (according to *Franklin and Paxinos, 2007*; *Kirkcaldie, 2012*) including core (black) and halo (grey) regions of mono-trans-synaptic anterograde tracer injections: primary motor (M1), primary somatosensory (S1), posterior parietal association cortex (PPC), primary visual (V1), primary auditory (A1), and dorsal auditory (AuD) cortex. Scale bar is 1 mm. (**B**) Schematic illustrating the topographical location of pontine labeling following injections. Colored lines indicate regions where labeled pontine neurons were found and dots (following A1

*Figure 3 continued on next page*

*Figure 3 continued*

injections) indicate regions where terminals were observed. (C) Percentage of mossy fiber labeling in the ipsilateral and contralateral cerebellum (left) and in its gross divisions (right): vermis, hemispheres and vestibulocerebellum (VbC). (D) Percentage of labeled mossy fiber terminals observed in the anatomical lobules (simplex lobule [Sim], crus I/II ansiform lobule [crus I/II], paramedian lobule [PM], copula pyramis [Cop], paraflocculus [PFl], flocculus [FL], vermal cerebellar lobules [I-X]) of the ipsilateral (left) and contralateral (right) cerebellum (% of the total cerebellar input).

terminals (~10,000) and AuD the least (~580; see *Supplementary file 1B*). Although the absolute number of mossy fiber terminals was variable, the resulting regional pattern of labeling was highly consistent within, and specific to, the cortical region injected (*Figure 3*). To examine the underlying organization of cerebrocerebellar projections, we quantified the proportion of mossy fiber terminals within each side (*Figure 3C*), division (*Figure 3C*), and lobule (*Figure 3D*) of the cerebellum.

Cerebrocerebellar projections were largely, but not exclusively, to the contralateral cerebellum (average across all animals: contralateral 70 ± 3%, ipsilateral 30 ± 3%). This was especially prominent after S1 injections, which resulted in 85 ± 3% of mossy fiber terminals in the contralateral cerebellum. However, this laterality varied across functionally distinct cerebrocerebellar input, such that the ratio of the number of mossy fiber terminals on each side was closer to equivalent for association/secondary cortical regions compared to that of primary cortical areas (ratio of ipsilateral: contralateral = 0.73 ± 0.13 versus 0.35 ± 0.06, respectively; p=0.008; two-sample t-test). Although M1 injections resulted in bilateral pontine labeling (*Figure 3B*, see also *Figure 1E*), the resulting mossy fiber terminals were still largely located in the contralateral cerebellum (74 ± 2%; *Figure 3D*, see also *Figure 1F*).

With respect to gross cerebellar divisions (*Figure 3C*), input from M1 and S1 was biased towards the cerebellar hemispheres (83 ± 3% and 55 ± 5%, respectively), with much smaller contributions to the vestibulocerebellum (8 ± 3% and 13 ± 3% respectively). The proportion of cortical input to the vermis was quite similar across cortical regions (~30%), with the exception of V1, which had very few vermal projections (14 ± 1%) and A1, which showed a comparatively strong bias towards vermal projections (47 ± 7%). Interestingly, projections from AuD deviated from A1 with a high proportion to the cerebellar hemispheres and without the vermal bias. Projections from V1 were biased towards the vestibulocerebellum (53 ± 5%) and PPC also showed a larger proportion of inputs to the vestibulocerebellum in comparison to other cortical areas. In fact, PPC was the only area that showed largely equal proportions of projections across the cerebellar subdivisions with no clear majority (hemispheres: 40 ± 3%, vermis: 30 ± 7%, vestibulocerebellum: 30 ± 3%).

We then quantified the proportion of mossy fiber terminals in each cerebellar lobule across both the ipsilateral and contralateral cerebellum (*Figure 3D*). After injections into M1, most resulting mossy fiber terminals were in contralateral crus II ansiform lobule (crus II), followed by the simple lobule (Sim), paramedian lobule (PM), and the crus I ansiform lobule (crus I). Although the absolute number of terminals from M1 was higher in the contralateral cerebellum, the relative proportion of labeling across the lobules was strikingly similar for both sides, resulting in a highly symmetrical pattern of labeling (*Figure 3D*, see also e.g. *Figure 1F*). Injections into S1 resulted in mossy fiber terminals largely in contralateral crus I, PM, and the copula pyramis (Cop). Indeed, S1 was the only cerebral cortical region with substantial input to the Cop (14 ± 5% contralateral projections; no other regions were >1.7%). Additionally, there was a relatively high proportion of terminals in contralateral vermal lobule IV/V after S1 injections. In contrast to M1 projections, S1 showed relatively few terminals in the Sim and crus II. Although the terminal pattern of S1 labeling was also largely symmetrical, the Cop was a clear exception, with only a small relative proportion of ipsilateral labeling (0.7 ± 0.3) compared to the contralateral side.

Injections in the association area, PPC, resulted in less laterality of terminal labeling within the cerebellum, as previously mentioned, however, there were substantial differences in the pattern of labeling between the ipsilateral and contralateral sides (*Figure 3D*). On the contralateral side, the relatively equal balance of mossy fiber projections to the cerebellar divisions is reflected in the high proportion of terminals to crus I (19 ± 6%; i.e. hemispheres), lobules IV-VI (10 ± 1%, i.e. vermis), and paraflocculus (PFl; 19 ± 6%, i.e. vestibulocerebellum). In contrast, the ipsilateral side showed less prevalent crus I labeling (4 ± 0.5%) but a relatively large proportion of projections to vermal lobule VI (13 ± 7%) and crus II (5 ± 2%). Therefore, cerebrocerebellar input from the PPC results in the lowest laterality, but also the least symmetrical, terminal organization.

With regard to the other sensory areas, after injections into V1, most labeled mossy fiber terminals were found in the contralateral PFl (44 ± 5%); while ipsilaterally the PFl also had a relatively high proportion of projections compared to other lobules (7 ± 2%), this remained less than a third of that to the contralateral PFl (*Figure 3D*). Crus I was the only region in the hemispheres with substantial terminal labeling from V1 (16 ± 6%, contralateral). Labeling in the vermis was sparse but highly symmetric, with lobule VI having the most prominent labeling (5 ± 2%), similar to M1 and PPC. Following A1 injections, a large proportion of the resulting mossy fiber terminals were found in the vermis (specifically lobules III [9 ± 5%], IV/V [12 ± 2%], and VI [8 ± 2%]), and in the cerebellar hemispheres there was a fairly equal distribution of terminals in crus I (8 ± 1%) and crus II (7 ± 1%). In general, A1 was the only cortical region with a similar proportion of projections specifically to the cerebellar hemispheres bilaterally (ipsilateral: 18 ± 2%; contralateral: 24 ± 2%). Finally, after injections into AuD, the most prominent mossy fiber labeling was in crus I (15 ± 3%) and crus II (8 ± 1%) followed by vermal lobules IV/V (7 ± 0.4%) and VI (7 ± 1%). In contrast to projections from A1, very few terminals were observed in lobule III (0.9 ± 0.1%). Although the absolute number of terminals from AuD was slightly higher in the contralateral cerebellum, the relative proportion of labeling across the lobules was strikingly similar bilaterally, resulting in a highly symmetrical pattern of labeling (*Figure 3D*).

## Lobule co-innervation of cerebrocerebellar inputs from distinct cortical areas

To determine the spatial overlap of cerebrocerebellar input from various cortical regions we examined the distribution of the total number of mossy fiber terminals within each lobule for each side of the cerebellum. All cerebellar lobules, on both ipsilateral and contralateral sides, received some cerebrocerebellar projections, however, the proportion of terminals in each lobule varied widely (ranging from 0.1–19.3% of the total ipsilateral terminal labeling and 0.1–16.9% of the total contralateral labeling; *Figure 4A,B*). The cerebellar hemispheres (Sim, crus I, crus II, PM, and Cop) received the majority of the total cerebrocerebellar mossy fiber input (68.2% of ipsilateral terminals, 67.1% of contralateral terminals), followed by vermal lobules IV/V and VI together (18.1% of ipsilateral terminals, 17.7% of contralateral terminals). Interestingly, we found a notable scarcity of mossy fiber terminals in posterior vermis, especially vermal lobule VII (1.2% of ipsilateral terminals, 0.4% contralateral terminals), which is contrary to studies of pontocerebellar projections alone (*Serapide et al., 1994*). Considering its relatively smaller volume, the PFl also received a large proportion of inputs (7.9% of ipsilateral terminals, 11.6% contralateral terminals; *Figure 4A,B*).

All cerebellar lobules also received mossy fiber inputs from two or more cortical regions; however, the proportion of cerebrocerebellar projections to a single lobule from each functionally distinct cortical region again varied substantially (*Figure 4A,B*). To examine the dominance of these projections from each target cortical region, we calculated the density of terminal labeling for each cerebellar subdivision according to the number of mossy fibers and the volume based on 16.4T MRT data (*Ullmann et al., 2012*; *Figure 4C*; *Supplementary file 1C*). In the cerebellar hemispheres, we found the highest density of terminals from motor input (M1), especially ipsilaterally, including Sim, crus I, crus II, and PM, but excluding the Cop (*Supplementary file 1C*). Bilaterally, the Cop had the highest density from somatosensory input; to the extent that 96% of the mossy fiber terminals found in the contralateral Cop originated in S1 (*Figure 4B*).

Lobules in the vermis had the highest density from S1 input (*Figure 4C*), except for ipsilateral lobule VI, which had a relatively large number of mossy fibers from M1 injections, and the most posterior vermal region, vermal lobules IX and X, which are part of the vestibulocerebellum (*Figure 4A,B*; see also *Figure 3D*). In general, the vestibulocerebellum had diverse cerebrocerebellar input. Bilaterally, the PFL and Fl had the highest density of terminals from S1 injections, although both also had substantial input from most other cortical regions (*Figure 4A–C*). Lobule IX and X also had sparse input from most cortical regions, however, IX had a proportionally large number of terminals from auditory regions (especially A1 but also AuD), while M1 was largely represented in lobule X (*Figure 4A,B*). The representation of inputs from A1 and AuD in vermal vestibulocerebellar regions is interesting considering the large number of labeled cells observed in the vestibular nuclei relative to other precerebellar nuclei from auditory cortical regions (see *Supplementary file 1B*). It is important to note, however, that most regions of the vestibulocerebellum (with the exception of the PFl) had a small density of mossy fiber terminals (*Figure 4C*, *Supplementary file 1C*); hence, while still representative, caution must be taken in interpreting the pattern of labeling within these regions.

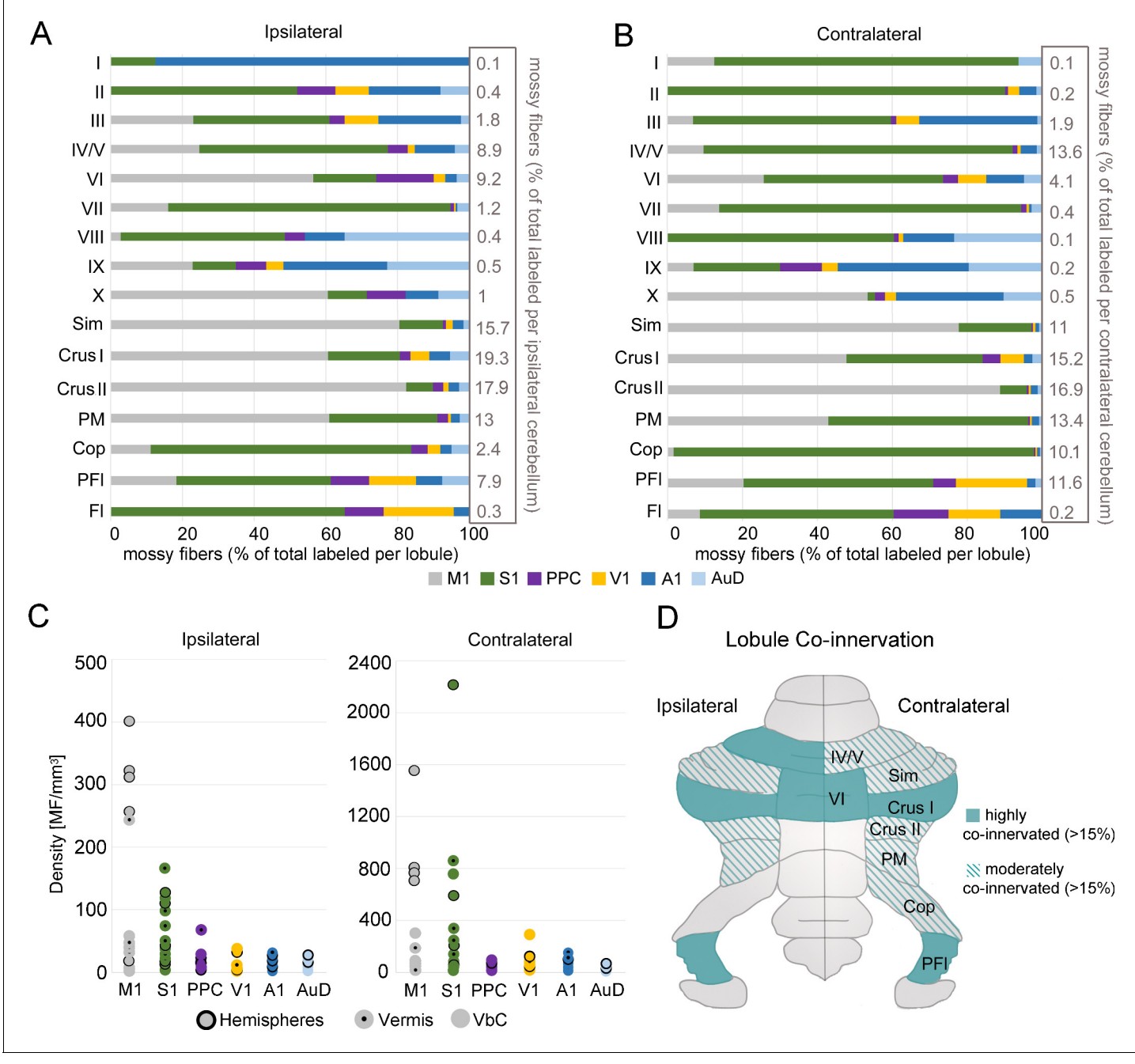

**Figure 4.** Regional convergence of cerebrocerebellar terminals originating across functionally distinct regions of cerebral cortex. (A–B) Proportion of labeled mossy fiber terminals in the ipsilateral (A) and contralateral (B) cerebellum following AAV injections into primary motor (M1), primary somatosensory (S1), posterior parietal association cortex (PPC), primary visual (V1), primary auditory (A1), or dorsal auditory (AuD) cortex. Values were normalized to the total number of labeled mossy fibers found within each lobule (simplex lobule [Sim], crus I/II ansiform lobule [crus I/II], paramedian lobule [PM], copula pyramis [Cop], paraflocculus [PFl], flocculus [FL], vermal lobules [I-X]) across all animals (right x-axis). (C) Density of mossy fiber (MF) terminals in the ipsilateral (left) and contralateral (right) cerebellum. Each dot represents the average density (number of MFs/mm$^3$) after each cortical injection M1, S1, PPC, V1, A1 and AuD. See also *Supplementary file 1C*. (D) Schematic of the unfolded mouse cerebellum summarizing the topography of mossy fiber inputs. Multimodal lobules receive at least 4% of the total mossy fiber input for that side (see A-B) and are highly (solid color) or moderately (striped color) multimodal based on input from at least two different functional cortical regions in addition to the dominant one (>15% and<15%, respectively).

We then classified the topography of mossy fiber inputs to the various cerebellar lobules according to the degree of regional overlap, as this would represent a prerequisite for potential multimodal processing between corticocerebellar inputs (*Figure 4D*). Since all cerebellar lobules received input from two or more cortical regions but some lobules had a low density of total mossy fiber terminals, we included subdivisions that received a minimum proportion of the total mossy fiber input for each side (>4%; see *Figure 4A,B*) and classified lobules as: 'highly co-innervated' if >15% of mossy fiber labeling was from at least two different functional cortical regions in addition to the dominant one, and 'moderately co-innervated' if this value was <15% (*Figure 4D*). Using these criteria, bilateral crus I, PFl, vermal lobule VI, and IV/V had the greatest potential for multimodal co-innervation; followed by bilateral Sim, crus II, PM, and contralateral Cop.

## Spatial organization of mossy fiber terminals in molecularly defined cerebellar modules

Cerebellar lobules contain highly organized parasagittally oriented modules based on anatomical, physiological and molecular subdivisions (for review see *Apps and Hawkes, 2009*). Therefore, to determine more precisely the potential for spatial overlap of multimodal information, we examined the distribution of mossy fiber terminals across the mediolateral and rostrocaudal extent of the identified multimodal lobules. To do this, we used the parasagittal expression pattern of aldolase C (or zebrin *Brochu et al., 1990*; *Ahn et al., 1994*) as a molecular marker, which is highly conserved across individuals. This allowed us to align the pattern of labeling observed in cerebellar regions across animals and, hence, quantify the spatial relationship between mossy fiber terminals originating from functionally distinct cortical areas (*Figure 5*, *Figure 6*, see also *Figure 5—figure supplement 1*, including validation of alignment across animals *Figure 5—figure supplement 1J*).

After alignment, for identified lobules with substantial co-innervation in the cerebellar hemispheres, terminals from both sensory and motor cortical areas were spatially overlapping (*Figure 5*) - with the exception of crus II, where M1 injections resulted in terminal labeling largely in dorsal regions and S1 in ventral regions (*Figure 5A*). Consequently, crus II had diverse functional cerebrocerebellar input but labeling was more spatially segregated within the lobule, with the distance between M1 terminals to themselves being significantly shorter than the distance between M1 terminals and those originating from other cortical injection sites (*Figure 5B*; M1-M1: 258 ± 22 µm, M1-Other: 305 ± 10 µm, p=0.007, two-sample t-test), this was also true for S1 (S1-S1: 190 ± 13 µm, S1-Other: 316 ± 15 µm, p<0.001, two-sample t-test). Hence, when identifying the nearest neighbors of M1 terminal locations in crus II, over 80% of these were also of M1 origin (*Figure 5C*); therefore, crus II has lower potential for multimodal spatial overlap at the modular level. In contrast, crus I showed highly spatially overlapping patterns, that is, the distance between both M1 and S1 terminal locations to that of other cortical origins was not significantly different to the distance between themselves (*Figure 5B*, M1–M1: 222 ± 8 µm, M1-Other: 220 ± 7 µm, p=0.953; S1-S1: 246 ± 10 µm, S1-Other: 233 ± 7 µm, p<0.001, two-sample t-test), and a higher proportion of their nearest neighbors came from extrinsic cortical origins (*Figure 5C*). Throughout the hemispheres, terminals from primary sensory regions V1 and A1 as well as association cortices (PPC and AuD) were generally located more on the apex of the folia, where they were intermingled with M1 terminals, whereas S1 terminals tended to be more at the base of the folia, also intermingled with M1 terminals.

This same apical/basal pattern was also observed in the PFl (*Figure 6A*); hence, the ventral PFl (part of the vestibulocerebellum) was highly co-innervated with terminal labeling from sensory and association cortices and the dorsal PFl largely contained S1 and M1 terminals. In contrast, in vermal lobules IV/V and VI terminals from all cortical regions were overlapping in more medial zones and S1 and M1 terminals were additionally in more paravermal zones, with little labeling from other modalities (*Figure 6*). Therefore, S1 and M1 terminals were quite widely distributed parasagittally, whereas terminals from other primary sensory and association areas tended to be in medial-vermal and lateral-hemispheric regions. In general, the pairwise distance between terminal locations was not significantly different between M1 and other cortical origins, however, S1 terminals were more tightly clustered to each other in both the PFl and lobule VI (*Figure 6B*).

Generally throughout the cerebellum, S1 injections resulted in the patchiest distribution with clusters of mossy fiber terminals in certain lobules aligning to the AldoC expression pattern; this was especially the case in Cop and to a lesser extent in the PM, where mossy fiber clusters aligned with AldoC+ stripes (*Figure 5—figure supplement 1D*). Interestingly, this pattern of projections is in

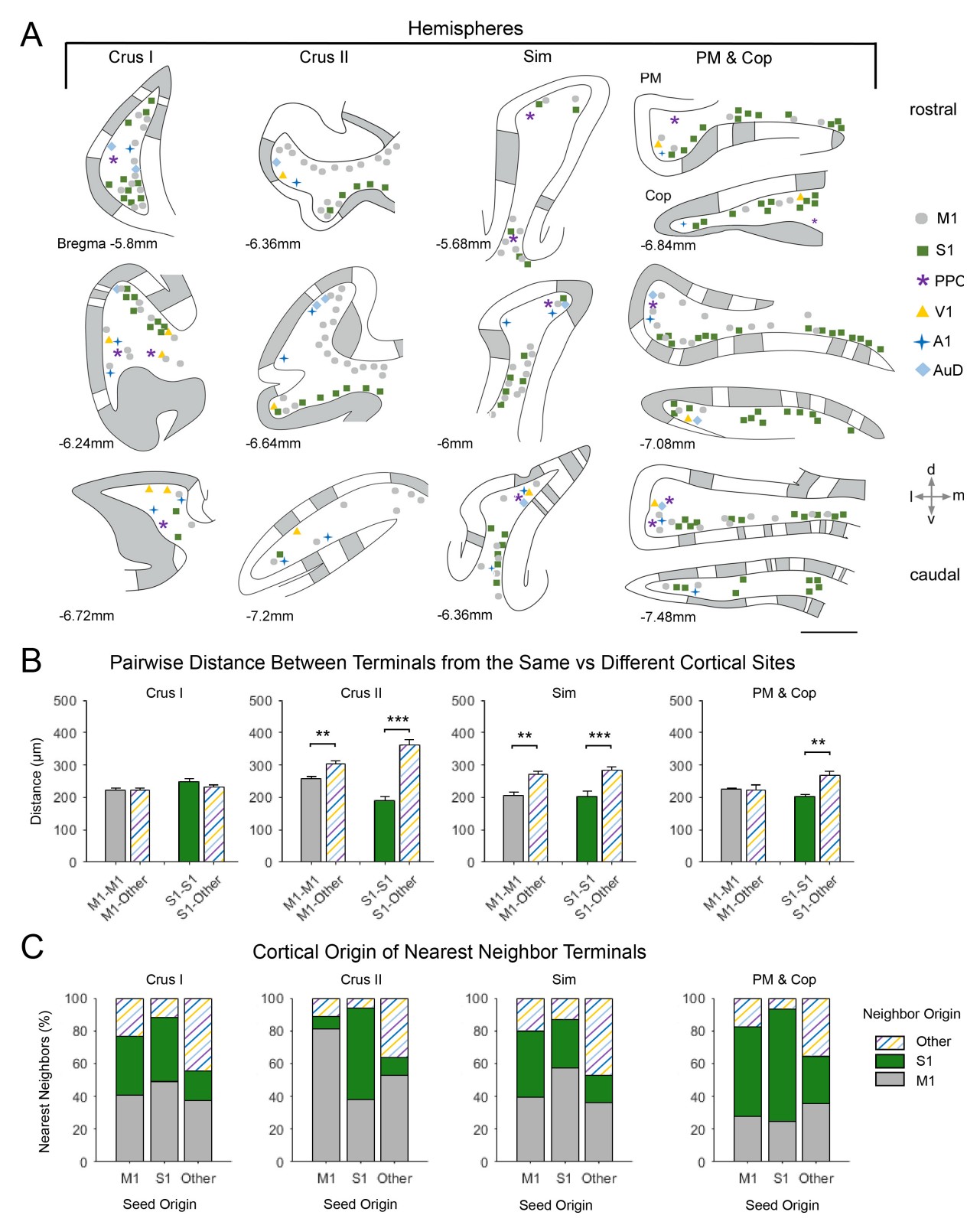

**Figure 5.** Spatial distribution of mossy fiber terminals within divisions of the cerebellar hemispheres with multimodal regional convergence. (A) Schematic reconstruction of the parasagittal distribution of labeled mossy fiber terminals in regions of the contralateral cerebellar hemispheres, after injections of mono-trans-synaptic AAV into target cortical regions: primary motor (M1), primary somatosensory (S1), posterior parietal association cortex (PPC), primary visual (V1), primary auditory (A1), or dorsal auditory (AuD) cortex. Data taken from 120 μm thick representative sections (three coronal

*Figure 5 continued on next page*

*Figure 5 continued*

sections) for rostral (top) to caudal (bottom) regions of the indicated lobules; distance from bregma is indicated based on the mouse stereotaxic atlas (*Franklin and Paxinos, 2007*). Data were aligned across animals using the parasagittal pattern of Purkinje cell molecular marker aldolase C (grey represents high expression, AldoC+; white represents low expression, AldoC-; see also *Figure 5—figure supplement 1*). For high density projections, M1 and S1, each dot represents 5–10 mossy fiber terminals, for all other lower density projections each dot represents 1–5 mossy fiber terminals. Simplex lobule (Sim), crus I/II ansiform lobule (crus I/II), paramedian lobule (PM), and copula pyramis (Cop). (B) Average pairwise distance between all terminals originating from the same cortical injection site (e.g. M1–M1) or from different cortical injection sites (e.g. M1-Other, where 'Other' is represented by all injected sites except M1 and S1). Statistical significance represents terminals being closer to like-terminals than to terminals from other cortical regions (**$p<0.01$, ***$p<0.001$, for exact p-values see *Supplementary file 1E*). (C) Proportion of the five nearest neighbors of each terminal location following each cortical injection site (i.e. seed origin: M1, S1 and Other; indicated on x-axis) that originate from M1, S1 or other cortical regions (i.e. neighbor origin: indicated on legend).

The online version of this article includes the following figure supplement(s) for figure 5:

**Figure supplement 1.** Parasagittal distribution of mossy fiber terminals and aldolase c (AldoC) expression.

agreement with results from electrophysiological recordings in the PM in rats, demonstrating alternating patches of somatosensory responses to hindlimb/forelimb stimulation (*Shambes et al., 1978*). However, distributed mossy fiber labeling was also present in both AldoC+ and AldoC- regions in the Cop and PM (*Figure 5—figure supplement 1D*), so that no specific bias was seen on average (*Figure 5—figure supplement 1G–I*). Conversely, in crus I clustered S1 mossy fiber terminals were more biased towards AldoC- regions (*Figure 5—figure supplement 1B,G*), although not strictly so.

To test for spatial randomness of terminal locations, we calculating the Ripley's K-function for each lobule and compared these distributions with simulated distributions of complete spatial randomness (CSR; see Materials and methods); for S1, all lobules except crus II ($p=0.260$) and lobule IV ($p=0.642$) showed significant deviation from CSR ($p\leq0.028$; $t = 60$ μm, 100 simulated CSR distributions; 3D Ripley's K-function; *Hansson et al., 2013*). Mossy fiber terminals from M1 injections generally had a more distributed, less clustered, pattern throughout the lobules and did not appear to consistently follow particular AldoC expression boundaries (*Figure 5*, *Figure 6*, *Figure 5—figure supplement 1E,F*); however, with the exception of lobule VI ($p=0.391$), M1 terminal locations still showed significant deviation from CSR ($p\leq0.015$; $t = 60$ μm, 100 simulated CSR distributions; 3D Ripley's K-function; *Hansson et al., 2013*). Injections in the other cortical regions also resulted in distributed and/or sparse terminal labeling which could not be assigned to a particular AldoC expression pattern and only deviated from CSR in crus I, sim, PM and Cop, and the PFl ($p\leq0.001$; $t = 60$ μm, 100 simulated CSR distributions; 3D Ripley's K-function; *Hansson et al., 2013*). Therefore, mono-trans-synaptic cerebrocerebellar mossy fiber terminals did not adhere to a specific pattern of zonal organization with respect to AldoC expression boundaries, however, many patterns of expression significantly deviated from spatial randomness. Since these terminals potentially relay in a number of different precerebellar nuclei (see *Supplementary file 1B* and *Figure 2*) our results do not preclude the existence of a finer-scale relationship with respect to individual cerebrocerebellar pathways and/or in a lobule specific manner.

## Intermediate cerebrocerebellar brainstem pathways

Based on the results of the mono-trans-synaptic tracer, there are a number of precerebellar nuclei, both pontine and extra-pontine, that have the potential to act as intermediate nuclei for cerebrocerebellar projections (summarized in *Figure 7A*; see also *Supplementary file 1B*). To gain insight into the likelihood that these various precerebellar nuclei act as intermediate sources of mossy fiber input to the lobules identified as having high co-innervation, we performed additional tracing experiments with injections of the retrograde tracer Cholera-toxin-B (CTB) into key vermal regions (lobules IV/V, IV) as well as cerebellar hemispheres (Sim and crus I; *Figure 7—figure supplement 1*). Additionally, we performed dual injections of CTB into lobule IV/V or crus I and the trans-mono-synaptic AAV1. cre into either M1 or S1, respectively (*Figure 7B*). Our results show that the majority of retrogradely labeled cells were located in the pons (across all cases: pontine = 6429 cells [73% of total]; extra-pontine = 2378 cells [27% of total]; *Figure 7—figure supplement 1*). With our dual injections, we also observed some double labeled cells in the pontine nuclei (e.g. *Figure 7B*), demonstrating direct

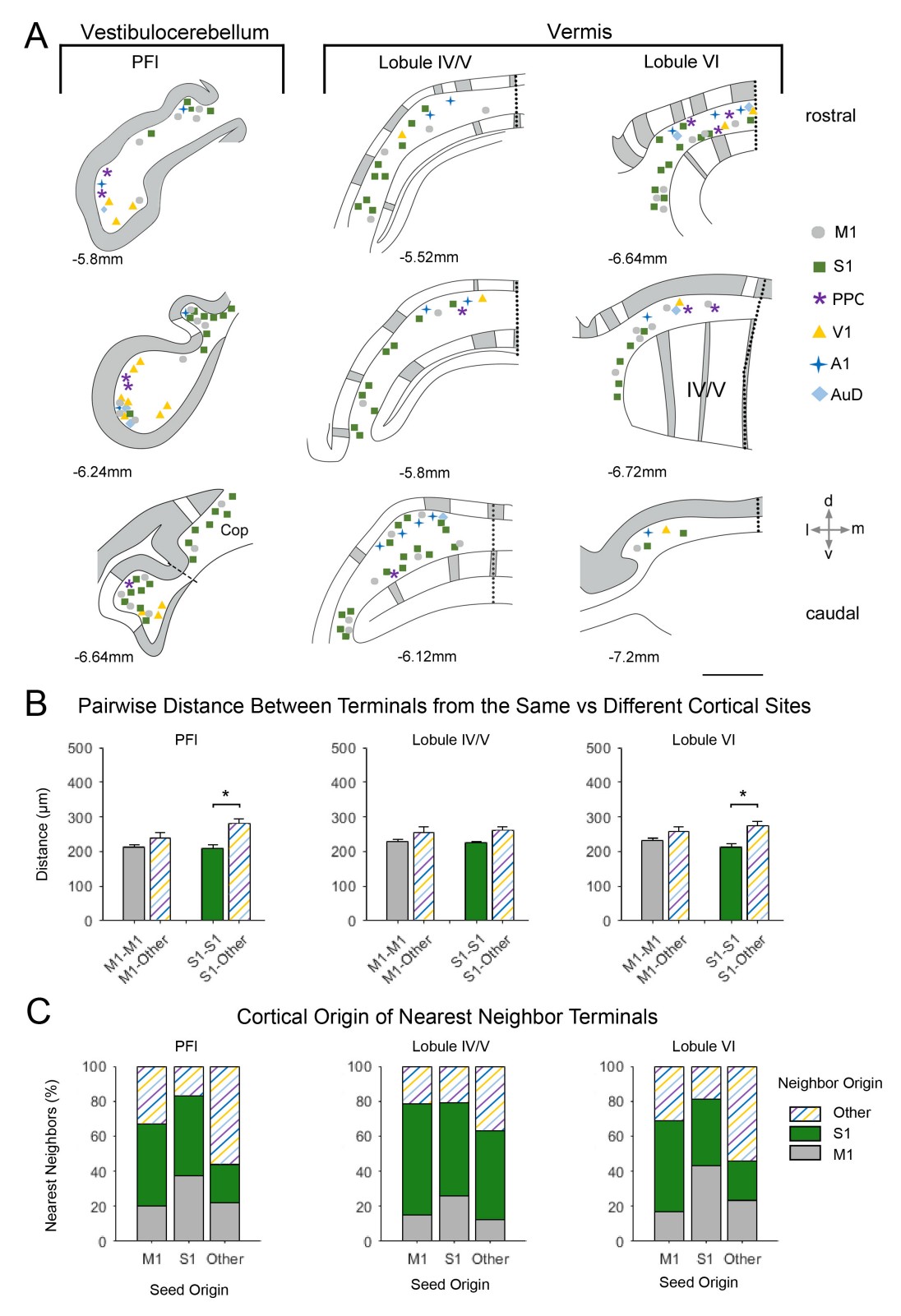

**Figure 6.** Spatial distribution of mossy fiber terminals within divisions of the vestibulocerebellum and vermis with multimodal regional convergence. (**A**) Schematic reconstruction of the parasagittal distribution of labeled mossy fiber terminals in contralateral cerebellar lobules spanning the vestibulocerebellum (VbC) and the vermis, after injections of mono-trans-synaptic AAV into target cortical regions: primary motor (M1), primary somatosensory (S1), posterior parietal association cortex (PPC), primary visual (V1), primary auditory (A1), or dorsal auditory (AuD) cortex. Data taken

*Figure 6 continued on next page*

*Figure 6 continued*

from 120 µm thick representative sections (three coronal sections) for rostral (top) to caudal (bottom) regions of the indicated lobules; distance from bregma is indicated based on the mouse stereotaxic atlas (*Franklin and Paxinos, 2007*). Data were aligned across animals using the parasagittal pattern of Purkinje cell molecular marker aldolase C (grey represents high expression, AldoC+; white represents low expression, AldoC-; see also *Figure 5—figure supplement 1*). For high density projections, M1 and S1, each dot represents 5–10 mossy fiber terminals, for all other lower density projections each dot represents 1–5 mossy fiber terminals. Dotted lines represent midline and dashed line represents division between paraflocculus (PFl) and copula pyramis (Cop). Vermal cerebellar lobules: lobule IV/V and lobule IV). (B) Average pairwise distance between all terminals originating from the same cortical injection site (e.g. M1–M1) or from different cortical injection sites (e.g. M1-Other, where 'Other' is represented by all injected sites except M1 and S1). Statistical significance represents terminals being closer to like-terminals than to terminals from other cortical regions (*p<0.05, for exact p-values see *Supplementary file 1E*). (C) Proportion of the five nearest neighbors of each terminal location following each cortical injection site (i.e. seed origin: M1, S1 and Other; indicated on x-axis) that originate from M1, S1 or other cortical regions (i.e. neighbor origin: indicated on legend).

confirmation that these pontine cells receive cerebrocerebellar disynaptic input. We note that the probability of double labeling was likely low due to the spatially restricted injection sites.

We also observed that many extra-pontine precerebellar nuclei identified as receiving cortical projections (*Figure 2*; *Supplementary file 1B*) contained retrograde labeling (e.g. lateral reticular nucleus, reticulotegmental nucleus, vestibular nuclei, interpolar part of the spinal trigeminal nucleus, and matrix region x; *Figure 7B*; *Figure 7—figure supplement 1*; *Supplementary file 1D*). Although double labeled cells were not directly observed in extra-pontine precerebellar nuclei, the labeling was spatially overlapping (*Figure 7B*), and quantification of the identity of the five nearest neighbors to cells originating from anterograde cortical injections, reveled that 13–45% of these were retrogradely labeled cells from the cerebellar cortex (*Figure 7C*; see also *Figure 7—figure supplement 2*).

From retrograde injections into the cerebellar cortex, we found that, while the majority of retrograde labeling was observed in the pontine nuclei, lobules IV/V and VI showed relatively higher proportions of extra-pontine retrograde labeling (IV/V, 44%; VI, 39%) in comparison to the cerebellar hemispheres (sim, 17%; crus 1, 21%; *Figure 7—figure supplement 1*). Lastly, although we did not observe significant cerebrocerebellar projections to lobule VII (e.g. see *Figure 3*), we confirmed that this lobule does receive pontocerebellar projections, but these were also proportionally few in comparison to other injected cerebellar regions (52% pontine vs 48% extra-pontine retrograde labeling, *Figure 7—figure supplement 3*), and originated largely from lateral regions of the pons, an area that topographically has been shown to receive corticopontine projections from the RSC (*Suzuki et al., 2012*; see also *Figure 1—figure supplement 1*).

Although, the proportion of observed mossy fiber terminals that relay through extra-pontine pathways cannot be precisely specified, we found a stronger positive correlation with the total number of mossy fiber terminals when all precerebellar cells are accounted for (R = 0.837, p<0.001, n = 18, Pearson correlation; *Figure 7D*) compared to when only pontine labeled cells are included (R = 0.632, p=0.012, n = 18, Pearson correlation). Additionally, we found lower variability in the total number of mossy fibers per labeled cell when all precerebellar cells are included (all cells: 50 ± 12 mossy fibers per cell; pontine cells only: 120 ± 63 mossy fibers per cell). Whether this ratio of ~50 mossy fiber terminals per precerebellar neuron can be generalized across individual cortico-precerebellar pathways remains to be determined, however, using single neuron tracing techniques, previous studies have reported a comparable 67 ± 7 (mice: *Biswas et al., 2019*) and 46 ± 9 (rats: *Na et al., 2019*) mossy fiber terminals in the cerebellum per pontine projecting neuron.

## Discussion

In this study we elucidated the cerebrocerebellar projections from primary motor, sensory, and secondary/association cortical areas to various regions of the mouse cerebellum. By using a mono-trans-synaptic anterogradely transported AAV (*Zingg et al., 2017*; *Zingg et al., 2020*), we found a highly organized topography of labeled corticopontine cells and also observed a number of potential extra-pontine cerebrocerebellar pathways. Notably, injections into A1 produced only terminal labeling in the pons, whereas injections into more dorsal secondary auditory cortex showed evidence of an auditory cortico-ponto-cerebellar pathway. We quantified the proportion and detailed the

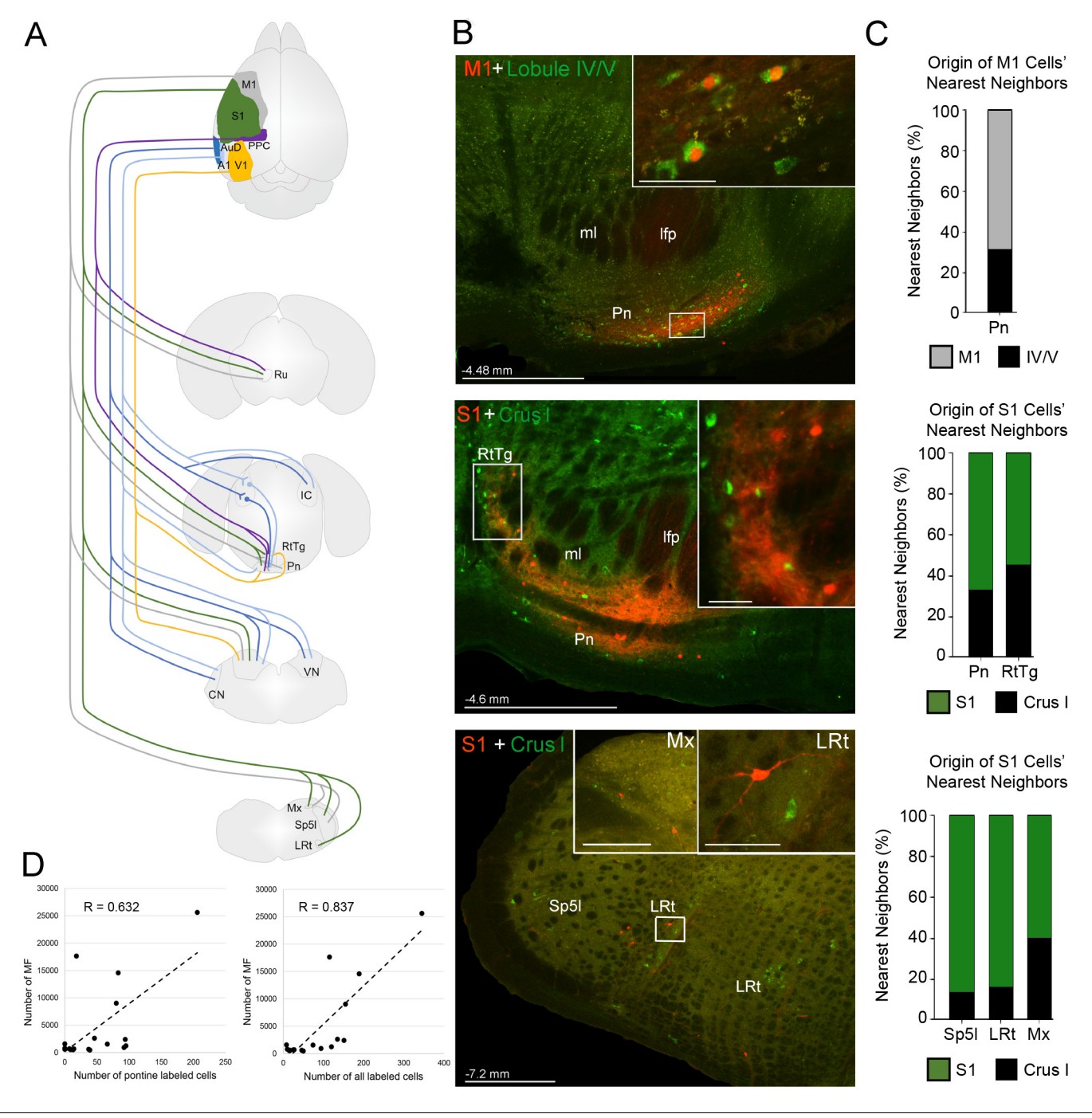

**Figure 7.** Summary of intermediate brainstem nuclei potentially supporting disynaptic cerebrocerebellar pathways. (**A**) Schematic summary of pathways to precerebellar nuclei from target regions of cerebral cortex: primary motor (M1), primary somatosensory (S1), posterior parietal association cortex (PPC), primary visual (V1), primary auditory (A1), or dorsal auditory (AuD) cortex. (**B**) Representative labeling in pontine (top) and precerebellar nuclei (middle, bottom) after combined injections of AAV1.cre (red) into cortical regions and CTB (green) into cerebellar regions (M1 + lobule lV/V; S1 + Crus I). Note the double labeled cells within the pontine nuclei (Pn; total 7 double labelled/52 anterogradely labeled cells from M1), and spatially overlapping cells within the reticulotegmental nucleus (RtTg) and lateral reticular nucleus (LRt). Distance from bregma is indicated based on the mouse stereotaxic atlas (*Franklin and Paxinos, 2007*). Scale bars 500 µm and 50 µm for insets. (**C**) Proportion of the five nearest neighboring cells for the cortical injection sites in (**B**) that are anterogradely labeled cells originating from the same cortical injection site (e.g. M1) or retrogradely labeled cells originating from injections in the cerebellar cortex (e.g. lobule IV/V; indicated by legend) within the indicated precerebellar nuclei. (**D**) Correlation of anterogradely labeled precerebellar cells with the total number of mossy fiber terminals per animal for either pontine only cells (left; R = 0.632, p=0.012, n = 18, Pearson's correlation) or pontine plus extra-pontine precerebellar cells (right; R = 0.837, p<0.001, n = 18, Pearson's correlation). Cochlear nuclei

*Figure 7 continued on next page*

*Figure 7 continued*

(CN), longitudinal fasciculus of the pons (lfp), matrix region x (Mx), medial lemniscus (ml), interpolar part of the spinal trigeminal nucleus (Sp5I), red nucleus (Ru), and vestibular nuclei (VN).

The online version of this article includes the following figure supplement(s) for figure 7:

**Figure supplement 1.** Retrograde tracing of cerebellar pathways from cerebral cortical receiving precerebellar target nuclei.

**Figure supplement 2.** Pairwise distance between anterogradely and retrogradely labeled cells within pontine (Pn) and extra-pontine precerebellar nuclei.

**Figure supplement 3.** Retrograde tracing of cerebellar pathways to lobule VII from cerebral cortical receiving precerebellar target nuclei.

precise organization of cerebrocerebellar terminals from each injected cerebral cortical region. The majority of labeled mossy fiber terminals from primary sensory and motor cortical regions were in the contralateral cerebellum, whereas projections from the secondary/association areas PPC and AuD resulted in less laterality in the cerebellum. Cerebellar subdivisions with the highest spatial overlap of multimodal cerebrocerebellar inputs within molecularly defined modules were bilateral crus I, PFl, vermal lobules VI, and lobule IV/V indicating that these regions may act as hubs for the integration of multimodal information originating from the cerebral cortex.

## Pontine and extra-pontine cerebrocerebellar pathways

Although the use of anterograde trans-synaptic tracers has not been extensive, our results in mice generally confirmed previous findings regarding the topographic organization of corticopontine pathways examined using anterogradely labeled terminals or retrograde pontine injections for mapping across other species; with more rostral cortical regions projecting more medially and caudal cortical areas projecting towards the lateral extent of the basal pontine nuclei (rats, *Odeh et al., 2005*; *Leergaard and Bjaalie, 2007*; cat, *Bjaalie et al., 1997*; monkey, *Glickstein et al., 1985*; *Schmahmann and Pandya, 1997*). Our results are also in agreement with the topographical organization of corticopontine anterograde terminal labeling in mice following S1 and M1 (*Proville et al., 2014*) as well as V1 (*Inoue et al., 1991*) injections. However, some differences may exist across species. For instance, although the current study did not compare various somatotopic regions within S1, following injections centered around the hindlimb/forelimb regions in mice, we did not find a substantial caudal bias in the pontine nuclei, as has previously been reported in rats from injections into similar cortical regions (*Leergaard et al., 2003*; *Odeh et al., 2005*); whether this reflects a species difference requires finer detailed somatotopic mapping in mouse S1 using these techniques.

Additionally, we found that the primary auditory cortex was unique in that it was the only cortical target region we observed with no direct corticopontine pathway. Following A1 injections, we found labeled mossy fibers bilaterally in the cerebellum (largely lobules III-V of vermis, PFl, crus I and II) but there were no intermediate cells trans-synaptically labeled in the pontine nuclei; indicating that this disynaptic mossy fiber input must reach the cerebellum through other precerebellar nuclei. In contrast, more dorsal auditory cortex (i.e. AuD) did have projections directly to the pontine nuclei. This is also consistent with our observations that mossy fiber terminals were more biased to vermal regions after A1 injections and included more projections to the cerebellar hemispheres (especially crus I) after AuD injections. There are substantial conflicting results in many species describing either the presence of an A1-pontine pathway (cats: *Perales et al., 2006*; rabbits: *Knowlton et al., 1993*; gerbils: *Budinger et al., 2000* and rats: *Wiesendanger and Wiesendanger, 1982*; *Legg et al., 1989*), while others found either no major A1-pontine projection or projections only from secondary/association auditory cortex (cats: *Brodal, 1972*; monkeys: *Glickstein et al., 1985*; rats: *Azizi et al., 1985*; bats: *Schuller et al., 1991a*), and studies in mice have not been extensive. However, through both anterograde and retrograde tracing, here we found no major A1-pontine pathway in mice.

After A1 injections, we did observe labeled *fibers* in the dorsolateral pontine nuclei, with apparent synaptic boutons - indicating a disynaptic functional connection with pontine cells. There are a few possibilities as to the origin of these fibers. First, they may be terminals labeled from trans-synaptically labeled cells in secondary auditory cortex; however, the pattern of pontine cell labeling we observed following AuD injections was topographically distinct from the location of these labeled fibers. Secondly, disynaptic A1-cerebellar pathways may travel via other extrapontine nuclei and only trisynaptically through the pons. For instance, here we confirmed direct projections from the IC to

dorsolateral regions of the pons (*Caicedo and Herbert, 1993*). Interestingly, anterograde labeling in the IC from auditory cortex was largely in the DCIC subregion, in which corticocollicular projections have been suggested to play a modulatory role on auditory processing (*Barnstedt et al., 2015*) compared to the sharper frequency tuning in CIC (*Stiebler and Ehret, 1985*; *Yu et al., 2005*). Further, A1 in mouse has been shown to be tonotopically organized, whereas the dorsal belt of auditory cortex is not (*Stiebler et al., 1997*; *Guo et al., 2012*), and auditory corticopontine projections in cats are also not tonotopically organized (*Perales et al., 2006*; whether this functional principle is a determinant for auditory corticopontine projections requires further consideration. Finally, another potential relay nucleus for these A1-pontine fibers is the cochlear nuclear complex, which is known to project directly to the pontine nuclei (*Kandler and Herbert, 1991*) as well as the cerebellum as mossy fibers (*Huang et al., 1982*; *Fu et al., 2011*). Thus, this precerebellar nucleus may serve as a disynaptic and potentially trisynaptic (*via* the pontine nuclei) auditory cerebrocerebellar intermediate, which may be functionally relevant in terms of temporal processing of auditory information. Future studies are needed to further delineate these polysynaptic cerebrocerebellar auditory pathways, including their functional significance in cerebellar processing.

Beyond corticopontine projections, we found a large - and in some cases equal - proportion of anterogradely labeled extra-pontine precerebellar cells in various brainstem nuclei, especially from S1, M1 and auditory regions. While the nature of the mono-trans-synaptic tracing does not permit the proportion of observed mossy fiber terminals that relay through pontine versus extra-pontine pathways to be precisely segregated, our observations from retrograde tracer injections into key cerebellar lobules, as well as evidence of observed fibers tracts travelling through the inferior cerebellar peduncle, suggest a number of precerebellar nuclei identified as receiving direct cortical input also have projections to cerebellar lobules with spatial overlap of multimodal cerebrocerebellar input. Therefore, it is likely that both pontine and extra-pontine nuclei act as sources of the observed mossy fiber input through disynaptic cerebrocerebellar pathways (see also *Watson and Apps, 2019*), and these precerebellar nuclei may play an important functional role to relay/integrate multimodal information from the cerebral cortex to the cerebellum (*Ruigrok, 2004*; *Fu et al., 2011*; *Sillitoe et al., 2019*). Since mossy fiber terminals were observed in the cerebellum but no pontine labeled cells were found after A1 injections, one can conclude that, at least for this disynaptic pathway, extra-pontine intermediate nuclei play a large role. The nature of relay versus integration of cortical signals along each of these precerebellar pathways will make for interesting future studies.

## Organization of cerebrocerebellar input to the cerebellum

Previous studies on pontocerebellar projections have found that the majority of mossy fibers terminate in the contralateral cerebellum (e.g. *Serapide et al., 2002*). Here we found that mono-transsynaptically labeled mossy fibers terminals from the cerebral cortex (through both pontine and extra-pontine pathways) project to a varying degree to the ipsilateral and contralateral cerebellum. Presumably these projections can cross either at the level of the pontine nuclei or the cerebellar peduncles, or both, as was recently found to be the case for some reconstructions of individual pontocerebellar neurons (*Biswas et al., 2019*; *Na et al., 2019*). We found that cerebrocerebellar pathways from sensory association and secondary cortices show less laterality than those from both primary sensory and motor cortices. Interestingly, using retrograde rabies tracing, *Suzuki et al., 2012* also found relatively large bilateral cerebrocerebellar pathways from orbitofrontal and retrosplenial cortical regions to the posterior cerebellar cortex. This pattern of laterality suggests that information coming from 'higher-order' prefrontal and cortical association areas may have a more global effect on cerebellar processing than more targeted 'unisensory' or primary motor inputs; whether this reflects a broader functional implication for the influence of the cerebral cortex in learning and predictive processing in the cerebellum is yet to be determined.

The cerebellar hemispheres as well as lobules VI and VII are traditionally described as cerebrocerebellar receiving areas (*Serapide et al., 1994*; *Kandel et al., 2000*); although see *Coffman et al., 2011*). We confirm the cerebellar hemispheres as the major target for cerebrocerebellar input in mice and detail the pattern of projections from primary motor and sensory areas, but also emphasize the diversity of disynaptic connections between the cerebral cortex and cerebellar cortex. Using this anterograde tracing technique, we found at least sparse cerebrocerebellar projections to every lobule of the cerebellum from motor and sensory cortices. Additionally, we found a relatively large number of labeled mossy fibers in lobules IV/V but few in more posterior vermal regions, including

lobule VII. Lobule IV/V is functionally considered part of the spinocerebellum and implicated in motor function, but not traditionally described as receiving extensive pontocerebellar or cortical input. However, recent studies have emphasized how this region is modulated by behavioral state and locomotive behaviors (*Jelitai et al., 2016*; *Muzzu et al., 2018*). Specifically, *Muzzu et al., 2018* suggested that predictive motor responses observed in mouse lobules IV/V may carry a motor efference copy coming from higher cortical areas. This is in agreement with our results, which confirmed disynaptic cerebrocerebellar as well as direct pontocerebellar input to lobule IV/V in mice. Conversely, our results of very few observed cerebrocerebellar terminals in lobule VII are seemingly in contradiction to previous findings of large pontocerebellar pathways to this region, which have been assumed to carry cerebrocerebellar information (*Päällysaho et al., 1991*; *Thielert and Thier, 1993*; *Biswas et al., 2019*). However, our results in mice are in agreement with the suggestion in other species that pontocerebellar inputs to lobule VII may relay cortical information of higher order than the primary motor and sensory areas, such as the retrosplenial cortex (rats, *Suzuki et al., 2012*) and prefrontal regions (rats, *Watson et al., 2009*; *Suzuki et al., 2012*; monkey, *Kelly and Strick, 2003*), or subcortical brain regions such as the superior colliculus and the accessory optic system (rats, *Mihailoff et al., 1989*; birds, *Pakan and Wylie, 2006*; monkeys, *Kralj-Hans et al., 2007*; for review see *Voogd and Barmack, 2006*).

## Functional implications

Our results show that cerebrocerebellar information originating from different functional cortical regions shows significant spatial overlap within molecularly defined modules in crus I, PFL, and vermal lobules IV/V and VI. Therefore, all gross divisions of the cerebellar cortex (hemispheres, vestibulocerebellum and vermis) have the potential for cortical multimodal influence, that is, the influence of the cerebral cortex is not strictly limited to the cerebellar hemispheres. These spatial overlaps constitute the fundamental anatomical basis for multimodal integration processes of cortical information at a modular and potentially cellular level (see also *Proville et al., 2014*). Functionally speaking, in vivo electrophysiological and imaging studies have shown that in rat (*Ishikawa et al., 2015*), mouse (*Chabrol et al., 2015*; *Giovannucci et al., 2017*; *Markwalter et al., 2019*) and mormyrid fish (*Sawtell, 2010*) single granule cells can respond to multiple modalities. For instance, *Ishikawa et al., 2015* found responses to sensory stimuli in half of the granular cells recorded in crus I/II and many showed responses to at least two different sensory modalities. Further, *Markwalter et al., 2019* also found that just over half of granular cells in vermal lobules VI respond to sensorimotor stimuli, with the majority exhibiting responses to multiple sources of distinct-modality stimulation. Although in functional studies examining granule cell integrative responses to date, the specific origin of mossy fiber inputs was either undetermined or originated from bottom up brainstem centers (e.g. *Chabrol et al., 2015*), it has also been shown that in decerebrated cats (*Jörntell and Ekerot, 2006*; *Bengtsson and Jörntell, 2009*) granule cells respond to only one modality. Therefore, the potential for multimodal convergence of cerebrocerebellar inputs in the granule cell layer that we have observed in this study may be an important determinant of granule cell function.

Alternatively, since the cerebellum receives both ascending and descending sensory and motor input, which has been shown to be anatomically integrated at the level of individual granule cells (*Huang et al., 2013*), it is also possible that the majority of basic sensory information is carried by ascending pathways and these descending cerebrocerebellar pathways are carrying *action-related* information from various cortical regions; including from primary sensory areas, in which a growing body of studies have demonstrated the influence of motor behaviors, such as locomotion, on sensory responses (*Niell and Stryker, 2010*; *Schneider et al., 2014*; *Pakan et al., 2016*; *Pakan et al., 2018*; *Ayaz et al., 2019*; *Poulet and Crochet, 2019*). In this way, the cerebellum may use these cortical signals in closed feedback loops to regulate and adjust ongoing predictive responses, as suggested by a feed forward model for motor control (*Kelly and Strick, 2003*; *Shadmehr et al., 2010*; *Gao et al., 2018*; *Chabrol et al., 2019*). Future studies examining the precise nature of the information carried by the cerebrocerebellar pathways identified here will further elucidate the functional influence of the cerebral cortex on cerebellar processing.

Assessing the full functional contribution of cerebrocerebellar input, and subsequent cortico-cerebellar loops, will require both advanced anatomical delineation of polysynaptic circuits combined with simultaneous observation and circuit manipulation in behaving animals. Mouse models, with

access to advanced genetic tools and the establishment of increasingly complex behavioral paradigms, provide many advantages in this regard. The results of the current study demonstrate conserved principles of organization between mice and other mammalian species, as well as provide foundational insight into the diversity, and potential for spatial multimodal convergence, within cerebrocerebellar pathways in mice. These findings will guide future research to delineate the precise nature of sensorimotor integration at multiple circuit levels.

# Materials and methods

## Key resources table

| Reagent type (species) or resource | Designation | Source or reference | Identifiers | Additional information |
|---|---|---|---|---|
| Strain, strain background *Mus musculus* | Strain: Ai9; Rosa26-CAG-LSL-tdTomato Cre reporter mice line Background:C57BL/6J | The Jackson Laboratory | RRID:IMSR_JAX:007909 | male and female |
| Recombinant DNA reagent | AAV1.hSyn.Cre.WPRE.hGH, | Addgene, | RRID:Addgene_105553 | titer 3.0 × 10 13 GC/ml |
| Recombinant DNA reagent | CAV.cre | PVM IGMM Montpellier, | | titer 5.5 × 10 12 GC/ml |
| Peptide, recombinant protein | Cholera-toxin-B 488 | ThermoFisher Scientific | Cat no. C34775 | |
| Antibody | anti-Aldolase C (H-11) (mouse monoclonal) | Santa Cruz Biotechnoloy Host species: mouse | RRID:AB_10659113 | 1:100 |
| Antibody | anti-RFP (polyclonal) | Rockland Host species: rabbit | RRID:AB_2209751 | 1:2000 |
| Antibody | Anti-Mouse IgG (H+L) Secondary, Alexa Fluor 488 Conjugated (polyclonal) | ThermoFisher Scientific Host species: donkey | RRID:AB_141607 | 1:200 |
| Antibody | Cy3-AffiniPure Goat Anti-Rabbit IgG (H+L) (polyclonal) | Jackson Immuno-Research Host species: goat | RRID:AB_2338006 | 1:200 |
| Software, algorithm | MATLAB 2013/2017a | Mathworks | RRID:SCR_001622 | |
| Software, algorithm | ImageJ (Fiji) | NIH – public domain | http://fiji.sc; RRID:SCR_002285 | |

Experiments were performed with 42 mice (9–12 weeks; both males and females) of the Rosa26-CAG-LSL-tdTomato Cre reporter mice line (Ai9; RRID:IMSR_JAX:007909). Note that this tdTomato reporter line is known to have increased levels of auto-fluorescence (*Hahn et al., 2019*), particularly in Purkinje cells (e.g. *Figure 1F*), however, this was easily distinguishable from cre-induced tdTomato expression under the microscope and in merged images. Animals were housed in standard laboratory cages (22℃, 12 hr light-dark cycle) with food and water available ad libitum. All experiments were performed according to the NIH Guide for the Care and Use of Laboratory animals (2011) and the Directive of the European Communities Parliament and Council on the protection of animals used for scientific purposes (2010/63/EU) and were approved by the animal care committee of Sachsen-Anhalt, Germany (42502-2-1479 DZNE).

## Neuroanatomical injections

For mono-trans-synaptic tracing of the cerebrocerebellar pathways we used an adeno-associated virus (AAV1.cre; AAV1.hSyn.Cre.WPRE.hGH, Addgene, USA, RRID:Addgene_105553) that expresses cre protein, which then drives tdTomato expression in the Ai9 reporter mouse line. The AAV1.cre is transported anterogradely to mono-synaptically connected neurons (*Zhao et al., 2017*; *Zingg et al., 2017*; *Zingg et al., 2020*) and subsequently also drives cre expression in anterograde target neurons (e.g. pontine cells); tdTomato reporter expression then fills these target neurons including axon terminals (e.g. mossy fiber terminals; *Figure 1*; see also *Zingg et al., 2017*). This AAV1.cre construct has been shown to spread exclusively to synaptically connected neurons with trans-synaptic spread

nearly abolished by tetanus toxin light chain (i.e. inhibition of presynaptic vesicle fusion; *Zingg et al., 2020*). Limitations in transport may exist through neuromodulatory projections (e.g. noradrenergic, cholinergic, serotonergic), however, efficient transport has been demonstrated for glutamatergic and GABAergic synapses to both excitatory and inhibitory neurons as well as long-distance projections (*Zingg et al., 2020*). Retrograde transport has also been shown with this AAV1.cre construct - although with lower efficiency (*Tervo et al., 2016*; *Zingg et al., 2017*). Therefore, using this technique, reciprocally connected regions may include cells that are both anterogradely and, to a lesser extent, retrogradely labeled; a clear advantage of using this technique to examine cerebrocerebellar projections in this study is that these pathways are largely descending and unidirectional (for review see *Léna and Popa, 2015*), which removes this as a confounding factor.

A total of 42 mice were used in this study. For mono-trans-synaptic tracer experiments, AAV1.cre injections (18 mice) were made into target cortical regions spanning multiple modalities, including primary sensory areas, primary motor regions and association cortex (3 mice per target): M1 (forelimb/hindlimb), S1 (forelimb/hindlimb), PPC (lateral and medial parietal association cortex), V1 (monocular), A1, and AuD. Further circuit investigations were performed as follows. An injection of CAV. cre (PVM IGMM Montpellier, France) was made into the pons (1 mouse) and an injection of AAV1. cre into the IC (2 mice). In a series of combined anterograde plus retrograde experiments (total 7 mice), injections of AAV1.cre into a target cortical region as well as injections of conjugated Cholera-toxin-B 488 (CTB; Thermo Fisher, USA) into individual cerebellar lobules were made (e.g. M1 + IV/V; S1 + crus I). Lastly, 14 mice were used for retrograde experiments alone, where conjugated CTB 488 (Thermo Fisher, USA) was injected into individual cerebellar lobules (IV/V, VI, VII, Sim and crus I) in order to examine the brainstem origin of mossy fiber projections to these key cerebellar regions.

All injections were performed as previously described (*Henschke et al., 2015*; *Henschke et al., 2018*). Briefly, animals were anesthetized with isoflurane (4%; Baxter, Germany), the cranial skin was incised, the skull exposed by a displacement of the skin and muscles, and a small hole was drilled into the skull. Cerebral cortical injections were always performed on the left side (in order to minimize potential variation due to lateralization effects), precluding the identification of lateralized projections. We used the following stereotaxic coordinates derived from the mouse brain atlas in reference to bregma (*Franklin and Paxinos, 2007*) and targeted cortical layer 5: M1: 1.7 mm rostral and 2 mm lateral; S1: −0.2 mm caudal and 2.2 mm lateral; PPC −2 mm caudal and 2 mm lateral; V1: −2.92 mm caudal and 2.5 mm lateral; A1: −2.8 mm caudal and 4.5 mm lateral; AuD: −2.3 mm caudal and 4 mm lateral. The following coordinates were used for the left IC: −5.02 mm caudal and 1 mm lateral, 1 mm deep. For all above injections, 60 nl of the AAV1.cre (titer $3.0 \times 10^{13}$ GC/ml) in $6 \times 10$ nl steps. The left side pontine nuclei were also injected: 6 mm caudal, 1 mm lateral and with an angle of 21°, 5.4 mm deep; with 100 nl of the CAV.cre (titer $5.5 \times 10^{12}$ GC/ml) in $10 \times 10$ nl steps. Since the majority of projections travel contralaterally from the cerebral cortex to the cerebellar cortex, we performed retrograde tracer injection on the right side cerebellar cortex targeting the granule cell layer: lobule IV/V: −5.9 mm caudal, 1 mm lateral and 0.7 mm deep; lobule VI: −6.6 mm caudal, 1 mm lateral and 0.5 mm deep; lobule VII: −7.6 mm caudal, 1 mm lateral and 1 mm deep; crus I: −6.8 mm caudal, 2.5 mm lateral and 1 mm deep; sim: −6.1 mm caudal; 2 mm lateral and 1.5 mm deep, with 200 nl of CTB in $10 \times 20$ nl steps.

All injections were made over 10 min into these areas *via* a nanoliter delivery system (World Precision Instruments, Germany) and fine glass micropipettes (tip diameter 20 μm). The speed of injection is important as trans-synaptic transport of the AAV1.cre construct is highly dependent on the titer within individual neurons (*Zingg et al., 2017*; *Zingg et al., 2020*); hence, small/slow volume injections where viral particles are not quickly dispersed through tissue were done to concentrate viral particles and optimize trans-synaptic activity. After injections, craniotomies were sealed with bone wax (Ethicon, Johnson and Johnson, Germany) and the skin was closed with tissue adhesive (Histoacryl; B/Braun, Germany). Subsequently, the animals were returned to their home cages for 2 weeks to allow for the expression of the viral constructs (both AAV1.cre and CAV.cre), and 5 days for experiments involving CTB injections alone. Note, for trans-synaptic labeling with the AAV1.cre construct, time periods longer than 2 weeks do not label additional orders of connections, that is labeling remains mono-trans-synaptic even over extended periods of time (*Zingg et al., 2017*).

## Histological processing

Animals were deeply anesthetized with ketamine (20 mg/100 g body weight, ip) and xylazine (1 mg/100 g body weight, ip) and perfused transcardially with 20 ml of 0.1 M phosphate-buffered saline (PBS, pH 7.4) followed by 200 ml of 4% paraformaldehyde. The brains were removed, post-fixed overnight in 4% paraformaldehyde at 4°C, and then cryoprotected in 30% sucrose in PBS for 48 hr. Brains were cut on a cryostat (CryoStar NX70, Thermo Scientific, USA) into 40 μm (cerebellum, brainstem) and 50 μm (cortex and underlying structures) thick coronal sections. Sections containing the cerebral cortex were directly mounted on gelatin-coated glass slides, sections containing the cerebellum were collected in PBS (free-floating) and blocked in normal donkey serum (10% and 0.4% triton in PBS) for 1 hr. Sections were then incubated in primary antibodies overnight at 4°C to visualize aldolase C expression (anti-AldoC; mouse, 1:100, Santa Cruz Biotechnology Cat# sc-271593, RRID: AB_10659113, USA) and red fluorescent protein (anti-RFP, rabbit, 1:2000, Rockland Cat# 600-401-379, RRID:AB_2209751, USA) for enhancement of the tdTomato signal. After rinsing in PBS, sections were incubated with the respective secondary antibodies (anti-mouse Alexa 488, 1:200, Thermo Fisher Scientific Cat# A-21202, RRID:AB_141607, USA; anti-rabbit Cy3, 1:200, Jackson ImmunoResearch Labs Cat# 111-165-144, RRID:AB_2338006, United Kingdom) for 2 hr. Finally, sections were rinsed again in PBS, mounted on gelatin-coated slides, and coverslipped with MOWIOL (Fluka, Germany).

## Quantification and data analysis

Sections of 40 μm thickness were examined using a confocal microscope (Zeiss LSM 700, Germany) equipped with a 2.5x objective (NA 0.085, Zeiss, Germany). For each section throughout the extent of the cerebellum two high-resolution images (2048 × 2048 Pixel) were acquired, one for each hemisphere. The two tiles were merged using inbuilt functions for feature based panoramic image stitching in MatLab (Mathworks, MA, USA; resulting in a ~ 4000×4000 pixel image of the entire cerebellum); brightness and contrast were adjusted as necessary using Adobe Photoshop software (v. 13.0.6 for Windows).

Only experimental cases where AAV1.cre injections were verified to be located within the aimed target areas M1 (forelimb/hindlimb regions), S1 (forelimb/hindlimb regions), PPC, V1, A1, and AuD were included in the analysis. The injection sites had a roughly cylindrical shape. All sections that covered the central core but not surrounding halo of each injection site were included in the analysis (see also *Figure 1—figure supplement 3*). We then calculated injection site volume as follows: $V = \pi * a * b * h$, where $h$ corresponds to the maximum value across sections for the height of the injection site [dorso-ventral direction], $a$ corresponds to the maximum value across sections for the radius of the injection site parallel to the cortical layers [medio-lateral direction], and $b$ corresponds to the rostrocaudal extent of the core injection site. The Line Measure Tool of the Zen software (Zeiss, Germany) was used to measure $h$ and $a$, and $b$ was calculated by counting the numbers of sections covering the core injection site multiplied by their thickness (40 μm).

The number of labeled mossy fiber terminals was evaluated by point marking mossy fiber rosettes in the reconstructed images, in all sections of each experimental animal using the Cell Counter plugin in ImageJ (Fiji, v. 1.43 r, NIH, Bethesda, MD, USA, RRID:SCR_002285). Counted mossy fiber terminals were assigned to either AldoC+/-cerebellar stripes using the ROI manager plugin of ImageJ. Unless otherwise stated, all data are reported as mean or, where applicable, mean ± standard error of the mean (s.e.m). The density $D$ of labeling for the ipsilateral as well as the contralateral side within each cerebellar lobule was calculated by taking the total number of mossy fiber terminals $N_{MF}$ for each side within each lobule and dividing by half the total volume $V$ of each lobule ($V_i$), in order to represent each hemisphere separately ($D = N_{MF}/(V_i * \frac{1}{2})$). Total volume measurements $V$ for each lobule were based on 16.4T MRT data (*Ullmann et al., 2012*; $V_I$ = 0.14 mm$^3$, $V_{II}$ = 1.16 mm$^3$, $V_{III}$ = 1.79 mm$^3$, $V_{IV/V}$=5.61mm$^3$, $V_{VI}$ = 2.5 mm$^3$, $V_{VII}$ = 0.67 mm$^3$, $V_{VIII}$ = 1.52 mm$^3$, $V_{IX}$ = 2.81 mm$^3$, $V_X$ = 1.26 mm$^3$, $V_{Sim}$ = 4.67 mm$^3$, $V_{CrusI}$ = 4.17 mm$^3$, $V_{CrusII}$ = 4.18 mm$^3$, $V_{PM}$ = 3.59 mm$^3$, $V_{Cop}$ = 2 mm$^3$, $V_{PFl}$ = 3.35 mm$^3$, $V_{Fl}$ = 0.79 mm$^3$).

For internal validation of AldoC+/-alignment across animals, we calculated Pearson's correlation for the frequency of mossy fiber terminals in AldoC+ and AldoC- regions of each lobule across animals. The pattern of labeling across lobules for animals with the same cortical injection sites (e.g. M1

- animal 1, 2, and 3) should be significantly correlated with each other if the AldoC alignment is consistent across animals.

For nearest neighbor analyses for both mossy fiber terminal locations as well as the location of anterogradely (from cortex) and retrogradely (from cerebellum) labeled cells, the five nearest events to each individual event (i.e. terminal location or labeled cell) was identified in turn and indexed according to their origin (i.e. cortical or cerebellar injection site) in order to quantify the proportion of nearest neighbors from the appropriate categories. Pairwise Euclidean distances between the identified nearest neighbours were calculated for each event and then averaged across events. Further, pairwise Euclidean distanced were calculated for all individual events in relation to all other events according to their origin (i.e. cortical or cerebellar injection site) and within a 500 µm radius.

Point-process analyses to test for spatial randomness of terminal locations were performed using a three-dimensional Ripley's K-function (*K(t)*; *Marani and Voogd, 1979*; *RipleyGUI*: *Hansson et al., 2013*), which can be simply stated as:

$$K(t) = \frac{E[Number\ of\ points\ within\ a\ distance\ t\ of\ an\ arbitraty\ point]}{The\ total\ point\ intensity}$$

in which a constant total point intensity is estimated using $\lambda = n/|V|$, where $n$ is the observed number of points in a region $V$ (*Jafari-Mamaghani et al., 2010*). Edge correction was employed (*Jafari-Mamaghani et al., 2010*) and a maximum distance of $t = 60$ µm was used. To quantify the deviation of a distribution from complete spatial randomness (CSR), a comparison set of 100 CSR distributions was created with the same properties (size and intensity) as the test distribution, the corresponding estimates of the K-functions are then processed to return a probability density function for each $t$. P-values were then acquired by locating $K(t)$ on the corresponding probability density function and calculating the area under the curve (*Hansson et al., 2013*).

## Acknowledgements

We are grateful to Doug Wylie and Eike Budinger for helpful comments on an earlier version of the manuscript as well as Cathleen Knape for expert technical assistance. This work was funded by the European Regional Development Fund (ERDF: Center for Behavioral Brain Sciences ZS/2016/04/78113).

## Additional information

### Funding

| Funder | Grant reference number | Author |
| --- | --- | --- |
| European Regional Development Fund | ZS/2016/04/78113 | Julia U Henschke<br>Janelle MP Pakan |

The funders had no role in study design, data collection and interpretation, or the decision to submit the work for publication.

### Author contributions

Julia U Henschke, Conceptualization, Data curation, Formal analysis, Validation, Investigation, Visualization, Methodology, Writing - original draft, Writing - review and editing; Janelle MP Pakan, Conceptualization, Supervision, Funding acquisition, Methodology, Writing - original draft, Project administration, Writing - review and editing

### Author ORCIDs

Julia U Henschke (iD) https://orcid.org/0000-0002-2435-8627
Janelle MP Pakan (iD) https://orcid.org/0000-0001-9384-8067

### Ethics

Animal experimentation: All experiments were performed according to the NIH Guide for the Care and Use of Laboratory animals (2011) and the Directive of the European Communities Parliament

and Council on the protection of animals used for scientific purposes (2010/63/EU) and were approved by the animal care committee of Sachsen-Anhalt, Germany (42502-2-1479 DZNE).

## Decision letter and Author response

Decision letter https://doi.org/10.7554/eLife.59148.sa1
Author response https://doi.org/10.7554/eLife.59148.sa2

## Additional files

### Supplementary files

• Supplementary file 1. Associated data quantification and Tables. (A) Table listing the location and size of mono-trans-synaptic adeno-associated virus (AAV1.cre) injection sites per animal. (B) Table showing the quantification of resulting anterograde labeling in precerebellar nuclei following injection of AAV1.cre into target cortical regions. (C) Table showing the density of mossy fibers within each cerebellar region following injection of AAV1.cre into target cortical regions. (D) Table showing the quantification of resulting retrograde labeling following injection of Cholera-toxin B retrograde tracer into target cerebellar regions. (E) Exact p-values associated with *Figure 5*, *Figure 6*, and *Figure 7—figure supplement 3*.

• Transparent reporting form

### Data availability

Data generated during this study (e.g cell counts, etc) are included in the manuscript and supporting files.

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
