## [Decision Letter]

**Acceptance summary:**

The authors have used a new mono-trans-synaptic tracing technique to characterize disynaptic cerebrocerebellar pathways. The results suggest a substantial amount of overlap within the cerebellar cortex of projections from distinct regions of cerebral cortex. The comprehensive results are beautifully presented and provide an important foundation for our understanding of how efference copies and other cortical information might be used by cerebellum to perform accurate sensory motor actions.

**Decision letter after peer review:**

[Editors’ note: the authors submitted for reconsideration following the decision after peer review. What follows is the decision letter after the first round of review.]

Thank you for submitting your work entitled "Convergence of multimodal inputs connecting the cerebral cortex to the cerebellum" for consideration by *eLife*. Your article has been reviewed by a Senior Editor, a Reviewing Editor, and three reviewers. The following individuals involved in review of your submission have agreed to reveal their identity: Richard Apps (Reviewer #2); Roy V Sillitoe (Reviewer #3).

Our decision has been reached after extensive consultation between the reviewers and the Reviewing Editor. Based on these discussions and the individual reviews below, we regret to inform you that we are not prepared to move forward with publication in *eLife* at this time. Although the reviewers agreed that the manuscript has the potential to serve as a thorough atlas for cortical projections to precerebellar nuclei, it was agreed that this goal was not achieved by the current manuscript. There were concerns regarding specific biological conclusions that can be reached at this stage, based on the data presented. However, the Editors and reviewers were enthusiastic about several aspects of this study. We have therefore prepared a summary of suggestions brought up during the consultation process, which involve new data and substantial revisions to the manuscript. Should these points, as well as the individual comments of the reviewers, be addressed in a reconfigured version of the paper, we would be willing to consider such a future submission for publication at *eLife*.

1) Overall, the reviewers felt that packaging this paper as a Short Report made it difficult to evaluate the findings, and we suggest expanding the manuscript to strengthen the evidence for the claims that are made. All reviewers wanted more documentation of injection sites (volumes and cell counts), and more evidence for truly comprehensive mapping. This was seen as critical for supporting the main potential strength of this paper as a comprehensive atlas study.

2) While all three reviewers supported the paper in principle, and agreed that it presents an interesting angle on topography and convergence of cortico-cerebellar connections, they each struggled with the data interpretation in light of limitations of the technique. The fundamental problem was that the experiments are unable to indicate the brainstem origin of the MF projection patterns. They may be mainly from the pons, but it is not possible to say if any of the other brainstem structures labelled are also sources. The reviewers came up with two suggested experiments to address this concern:

a) In the spirit of an "atlas paper," the authors could confirm pontine projections to the labeled and unlabelled (A1) regions of cerebellum using retrograde AAV (to rule out extrapontine projections). This could validate the method and strengthen novel findings such as pontine projections to lobes IV/V.

b) Another possibility could be to do dual viral injections: (transsynaptic) AAV-cre in cortex, and AAV-floxed-XFP in pons. If feasible, this could confirm the presence of certain cortical-pontine-cerebellar projections.

3) It was agreed that the conclusions regarding potential multimodal integration in individual granule cells were not adequately supported by the data. Rather, this was seen as one possible (and reasonable) interpretation based on a general overlap of MF labelling. Although reviewers (particularly #1) suggested specific experiments that could strengthen this claim, we do not think that it is critical to demonstrate this kind of convergence in the current manuscript. The demonstration of broad overlap itself is interesting. However, the conclusions regarding ultimate convergence should be toned down. The current findings can be presented as broadly consistent with the idea of multimodal integration in individual granule cells, which has been previously suggested by earlier studies (eg Huang, Ishikawa…).

We hope that you will find these comments and the reviews below helpful in preparing a revised version of the manuscript.

Reviewer #1:

In this manuscript, Henschke and Pakan describe a beautiful set of experiments in which they used an AAV2.1-based anterograde, fluorescence protein-based tracing technique (Zingg et al., 2017) to elucidate cerebro-pontine-cerebellar pathways. They focused on primary sensory (S1, V1 and A1), motor and one association area (PPC). They observed, as in previous studies, that these regions target ipsilateral basal pons in a spatially segregated manner. In turn, cerebellar mossy fiber (MFs) are found to be labelled in specific regions cerebellum including both in the vermis and lateral hemispheres. These results are consistent with other tracer and electrophysiological studies. Interestingly they find that A1 does not provide pontine-cerebellar projections, but dorsal auditory cortex does. This contrasts previous results in other species. Finally, and most novel, the authors examined the spatial overlap of MF projections from different cortical regions and show that in some regions multimodal processing is likely. While this has been shown for single granule cells in vivo in Crus I (Ishikawa et al., 2016), a survey of what cerebellar regions are likely to receive multimodal cerebro-pontine input had not been shown previously. In summary, the authors provide provoking evidence of a strategy to specifically label and manipulate different that pontine-cerebellar circuits will be possible.

The technique and results described in this manuscript are illuminating, and the data thoroughly analyzed. The manuscript, therefore, has the potential to serve as a thorough atlas for cortical projections to precerebellar nuclei. This study paves the way for future studies to refine our understanding of how efference copies and other cortical information might be used by cerebellum to perform accurate sensory motor actions. Unfortunately, limitations in the method preclude a clear biological conclusion from the current dataset, beyond the notion that cortical-pontine-cerebellar wiring is similar in mice as other species:

1) The authors show that AAV1-cre injections in most cortical regions tested indeed label the pons, but also extra-pontine precerebellar nuclei. This highlights the fundamental problem that we cannot be sure that the MF projection patterns in cerebellum are solely a result of pontine projections -as implied in the manuscript. One example of a new biological conclusion is that cerebro-pontine projections were largely thought to be restricted to posterior and lateral cerebellum, but here the authors seem to show prominent projections in lobes 4 and 5. However, we cannot be sure of this conclusion because the MF labelling in lobes 4 and 5 could be from another precerebellar nucleus. One possible experiment to confirm that pons projects to a given region of cerebellum would be to perform retrograde labelling from different cerebellar regions identified in the anterograde transynaptic method, and verify that cell bodies in pons are labeled.

2) Another conclusion proposed by the authors is that multimodal integration is likely in certain cerebellar regions. While overlapping domains of MF projections is necessary to achieve multimodal integration. A critical question is whether single granule cells receive different cortical inputs. Such multimodal combinations have been described for motor and proprioception (Huang et al., 2015), for visual, somatosensory and auditory (Ishikawa et al., 2015), as well as visual and vestibular information (Chabrol et al., 2015). Understanding the role of multimodal processing in cerebellum requires knowing whether cortical information is "mixed" at the level of single granule cells, as proposed by Marr and Albus. Are there specific wiring rules for cerebro-ponto-cerebellar projections? For example, perhaps corollary information only pairs with its principal sensory modality partner. Unfortunately, these experiments are more challenging and would require two-color labelling using another conditional method (e.g. Flp), in addition to a strategy to sparsely label GCs (electroporation, patch or mouse lines (Huang et al., 2013)). Nevertheless, this would significantly advance our understanding of how cortical information is processed at the input layer of cerebellum.

3) The lack of cortico-pontine projections into cerebellum from A1 is surprising, in light of previous work. Thus, it is necessary to independently confirm this result using retrograde labelling strategies. If true, a better discussion of why it might be different than previous results is merited. Finally, another concern about the A1 experiments, is that in Figure 1—figure supplement1 AAV1-cre injections there is very little labeling of inferior colliculus. As I understand the method, retrograde labelling is also expected. As the exact mechanism of transynaptic propagation of the virus or Cre is not known, is it possible that the method failed for this brain region?

I also have additional specific concerns.

1) When examining table 2, I noticed that similar numbers of labelled precerebellar cell bodies can produce differences in the number of MFs by nearly a factor of 10. Does this discrepancy limit the ability normalize for labelling efficiency and thus limit quantification of the results? How could this happen?

2) The conditions used to perform imaging should be better described. For example, how was high-resolution imaging performed with a 2.5 x objective? What was the NA of this objective? Given the accuracy of the galvanometer mirrors, necessary zooms may not be possible to achieve true diffraction limit resolution. The pixel and image sizes should be mentioned.

3) How was the injection volume measured? The width of the column size measure was presumably estimated from a line profile along an image, but what width parameter was taken as a width. The images shown are clearly saturated. Is this the same image in which the size of the infected region was estimated?

Reviewer #2:

This is a short report using a trans mono synaptic anterograde tracer method to chart cerebro-cerebellar anatomical projections in mice. Injections into a number of different neocortical regions (M1, S1, V1, A1, posterior parietal association cortex or the dorsal field of auditory cortex) revealed topographically organized projections to the pons and mossy fibre terminals in the cerebellar cortex. The report is generally well written and provides a valuable 'road map' for future studies of the functional significance of cerebro-cerebellar projections in mice.

Essential revisions:

1) The report needs to make clear what are the novel findings.

2) How did differences in cell counts and other measures of injection site affect the pattern and extent of anterogradely labelled cell and mossy fibre terminals? It was not clear to me if Figure 2A shows the total extent of injection sites into different neocortical areas or just the defined regions of interest. Either way the area clearly varies considerably between targets which will affect projection density and possibly topography.

3) The histogram axes in Figure 2D are not all drawn on the same scale making comparisons difficult. Also percentage values are potentially misleading given the large variation in cell counts and mossy fibre terminals listed in Figure 1—figure supplement 1B.

4) Given that the pons is the primary source of mossy fibre inputs, how did the authors determine that other brainstem nuclei also containing labelled cells in the same experiments necessarily provided mossy fibre projections?

5) The topographical organization within the pons should be compared to the principles of organization defined in previous studies (eg Leergaard, 2003; Odeh et al., 2005).

6) The discussion should consider the extent to which cerebro-cerebellar pathways are conserved across mammalian species. How useful is the mouse as a model for study of cerebro-cerebellar projections?

Reviewer #3:

Henschke and Pakan have submitted a manuscript entitled "Convergence of multimodal inputs connecting the cerebral cortex to the cerebellum" in which they use modern neuronal tracing methods to map circuit projections into the mouse cerebellum. The authors traced the cerebellar inputs originating from sensory, motor, and association cortices. They argue that diverse functional connections project from the cortex to cerebellum, where they ultimately exhibit a convergent terminal field distribution within specific lobules. The manuscript is well written and technically, it offers a much-needed approach to better understand the organization and trajectories of major cortico-cerebellar projections. In addition, the figures are well made, schematics very helpful, and overall the panels are easy to follow. However, below we provide a number of comments and concerns that we hope will improve the clarity and impact of the study.

1) Throughout the text, the discussion of A1 and its unique projection through non-pontine pre-cerebellar nuclei is often highlighted, however the significance of this point is not well articulated. Further clarification in the abstract and in the body is needed if the authors wish to highlight the uniqueness of this pathway. This is especially true given the findings in the text that S1 and M1 both have substantial, if not even majority in the case of S1, projections to non-pontine pre-cerebellar nuclei.

2) Introduction. Please clarify why the identified cortical regions were selected. In addition, while we agree that the characterization of the injections is rather comprehensive, there is a limitation in this study in that either the number of cortical regions chosen or the breadth of the characterization of the selected regions (ie: why weren't at least two neighboring sites in a given cortical region analyzed) prevent the present study from providing a fully comprehensive map, as stated by the authors.

3) Figure 1A. Please indicate the circumscribed subregion in each cortical area where the injection was made.

4) Subsection “Anterograde tracing of indirect cerebrocerebellar pathways”. Please clarify percentages for A1 projections in order to make the second sentence "This pattern of extra-pontine labeling was similar…" clearer. Further, in Figure 2B, while the scarcity of the projections to the pontine nuclei is similar, they seem to be qualitatively different in that A1 projects terminals, while the AuD experiment labels cell bodies. This distinction is not clear in the text. Moreover, the fact you see terminals in one region and then cell bodies another makes it a challenge to piece together what the actual pathway(s) might look like. That is, how do you substantiate the claim that the cell bodies in the pons (the unexpected labeling) actually project to the cerebellum? And on the flip side, for the majority of S1 labeling, which is in the precerebellar non-pontine nuclei, can we really assume that these all project to cerebellum? If not, what percentage?

5) The Supplementary file 1 is a bit difficult to interpret. Please include the percentages referenced in the text in the table. In addition, it is unclear why in some injection sites the sum of the individual pre-cerebellar nuclei cells equals the summed total, while in other cases the sum is less than or greater than the listed total. (ie M1-1 52 vs 55; V1-1 8 vs 8; A1-1 10 vs 8)

6) The significance of terminals in pontine/pre-cerebellar nuclei without cell body labelling is unclear and there should be an explanation of this in the text. For instance, in regard to A1/AuD does this indicate inefficiency in trans-synaptic labelling, that there is an alternative monosynaptic projection from A1/AuD to the pontine nuclei, or is there an alternative explanation? This is mentioned for M1 projections to the olivary nuclei (Subsection “Anterograde tracing of indirect cerebrocerebellar pathways”).

7) In the subsection "Convergence of multimodal cerebrocerebellar input within lobules…" the metric for dominant representation (> or < 55%) is problematic given the relative sparsity of the injections (either across or within cortical regions). Furthermore, given the variability of numbered terminals across individual animals in a given cortical area (ie from 2406 to 25616 from the M1 injections), this type of quantification and conclusions about convergence seem premature.

8) While the discussion about convergence and integration is quite interesting to speculate upon, it is hard to grasp whether the current data is sufficient to reach such a conclusion. Given the vast number of granule cells and the fine scale nature of cerebellar microzones, conclusions about integration of multimodal afferent information by single units (whether microzone or granule cell) cannot be easily drawn. To do this, a method distinct from the approach used in this paper would be needed whereby labelling from two distinct cortical areas could be distinguished in the cerebellum in a single animal. In the end, I am having trouble fully appreciating how integrated the "multimodal" and "convergence" nature of the pathways actually is, based on the data provided.

9) Finally, there is a core assumption at play, which is that cortical injection - pontine/precerebellar cell body labeling - mossy fiber terminal marking. In the case of cortico-ponto-cerebellar pathways, the abundance of literature makes this assumption sound. However, in the case of the precerebellar projections (especially S1 which has a majority non-pontine projections), sagittal sections (for example) demonstrating the axonal projections, and elucidating the monosynaptic pathway would provide a helpful piece of evidence to support the current content.

1) In the Abstract, it is not clear what is actually meant by "Prominent terminal overlap across motor and sensory modalities…". And, how was this measured?

2) It would be very helpful to include the species studied in the Title and/or the Abstract.

3) It is critical that the authors fully outline the limitations of the mono-trans synaptic anterograde viral tracer, in their hands, and specifically for the purpose of this study.

4) In Figure 1B, it is not clear what the red arrows are representing and pointing towards. Aren't the arrows pointing the wrong way if this is to show anterograde tracing direction?

5) Is there a time dependence on the tracing? That is, do longer periods of survival after injection label additional projections?

6) In Figure 1F, M1 should project ipsilateral, but in the cerebellum one can see many terminals on both sides. Please explain.

7) The authors state "…extent M1, indicating the presence of a polysynaptic cerebro-olivary pathway…" What is the literature to support this?

8) The authors state "We quantified the number of resulting mossy fiber terminals throughout the extent of the cerebellum and found that this was highly variable across cortical regions…" How variable is the size of the infected cell population in the cortex?

9) The authors state "There was a clear and consistent topography of…" It would be helpful to fully define what you mean by topography here. There are many levels of topography, in different brain regions, and this could mean very different things to different researchers.

10) In Figure 3E, what does each data point represent? A single terminal?

11) Along the same lines as above, it would be helpful to see more actual data with the combined zebrin labeling. Mossy fiber patterns, by nature of their clustered appearance, are not so easily appreciated and therefore additional examples, at higher power magnification, would be welcomed.

12) The authors state "This allowed for a detailed comparison of cerebellar regions across brains and, hence, across functional cortical regions representing multiple modalities (see Figure 3—figure supplement 1)." However, I don't see a full description or explanation of the data in the text.

13) The number of animals used in the study seems quite low.

14) In Figure 3—figure supplement 1, what is the red staining in Purkinje cells?

[Editors’ note: further revisions were suggested prior to acceptance, as described below.]

Thank you for resubmitting your work entitled "Cerebral cortical connections to the cerebellum across multiple modalities" for further consideration by *eLife*. Your revised article has been evaluated by Richard Ivry (Senior Editor) and a Reviewing Editor.

The manuscript has been improved but there are some remaining issues that need to be addressed before acceptance, as outlined below:

Summary:

Henschke and Pakan have provided a substantially revised version of a paper describing the connectivity between different areas of the cerebral cortex with the cerebellum. The authors have used a virus based transsynaptic tracing approach to show that multi-modal inputs may converge upon specific topographical domains within the cerebellar vermis and hemispheres. The study is anatomical in nature and takes an in-depth analysis of mossy fiber terminal field distributions in the cerebellum, but the authors also examine the different intermediate projections of the cerebral-cerebellar projections that are located within various brainstem nuclei. The paper is well-written and the images are high quality. reviewers particularly appreciated the addition of important controls (characterization of extra-pontine sources of MFs, confirmation of pontine sources using retrograde labeling, and identification of an A1-colliculo-cerebellar pathway), which significantly increase confidence in their original findings and propel the study towards the authors' original objective as a seminal reference, or atlas, for the characterization of cerebrocerebellar mossy fiber (MF) pathways.

Essential revisions:

The major remaining concern – shared by both reviewers – is the overemphasis/ speculation on 'multimodality' for this essentially anatomical study. Specifically:

1) I was delighted to see a more detailed examination of the location of MFs from different modalities within each lobule, since a prerequisite for multimodal processing is that the information project to the same subregion of a lobule. I also found rather striking the case in Crus II where there was still segregation between two MF types within the same lobe. This latter result highlights the notion that multimodal innervation within lobes is necessary but not sufficient to conclude multimodal processing. I therefore encourage the authors to soften their choice of words of "highly multimodal" and "moderately multimodal" as a conclusion of Figure 4. Perhaps the use of the term "lobule co-innervation" is more appropriate. Then after Figure 5, the argument for putative multimodal processing is much stronger, but still only suggestive.

2) Indeed, Figure 5 shows an essential set of analyses suggestive of multimodal processing, but I was left wanting for some clear quantification of "intermingled" and a simple overall summary.

a) Would it be possible to analyze the distribution of nearest-neighbor distances between terminals of different origins? Those that co-innervate would have shorter distances and those that do not, e.g. Crus II, would show a shifted distribution. Perhaps a more sophisticated point-process analysis, such as the application of the bivariate Ripley K-function, could be used to test for spatial randomness between two species of points and provide statistical confidence.

b) To add some statistical rigor to the alignment with adolase C staining, one could compare the frequency of AldoC positive and negative terminals for each cortical injection. Statistical tests for the frequency of events can then be used.

3) The title continues to hint at a more functional set of analyses, which the paper really does not provide. Please revise.

4) Similarly, the Abstract could present the key anatomical distinctions that were uncovered rather than the speculative discussion about multimodal sensory input – this functional idea should remain, but it should be toned down to make space to better clarify the actual anatomical findings.

5) The main Figure 6 merits a more thorough quantification of retrograde labeling.

6) Quantification and presentation of co-labeled and spatially intermingled neurons following dual viral tracing injections would be appropriate.

7) I like the estimation of the number of MFs per pontine neuron and would like to see it as a main figure.

8) The Discussion section rambles on a bit. I believe it can be shortened and used to more clearly summarize the major findings, the similarities and differences with previous results, and the implications for cerebellar function.

9) The section titles are rather vague. For example, "Organization of cerebrocerebellar mossy fiber terminals" is essentially duplicated in the Results and Discussion, and perhaps needs more specific wording. "Spatial organization of cerebrocerebellar mossy fiber types". I am not really sure why there is a Discussion subsection on "Diversity of cerebrocerebellar pathways" as you define the pathways by viral injection. Perhaps something like: "Cerebellar cortical target diversity of cerebrocerebellar pathways?"

---

## [Author Response]

[Editors’ note: the authors resubmitted a revised version of the paper for consideration. What follows is the authors’ response to the first round of review.]

Our decision has been reached after extensive consultation between the reviewers and the Reviewing Editor. Based on these discussions and the individual reviews below, we regret to inform you that we are not prepared to move forward with publication in eLife at this time. Although the reviewers agreed that the manuscript has the potential to serve as a thorough atlas for cortical projections to precerebellar nuclei, it was agreed that this goal was not achieved by the current manuscript. There were concerns regarding specific biological conclusions that can be reached at this stage, based on the data presented. However, the Editors and reviewers were enthusiastic about several aspects of this study. We have therefore prepared a summary of suggestions brought up during the consultation process, which involve new data and substantial revisions to the manuscript that are likely to take more than two months. Should these points, as well as the individual comments of the reviewers, be addressed in a reconfigured version of the paper, we would be willing to consider such a future submission for publication at eLife.1) Overall, the reviewers felt that packaging this paper as a Short Report made it difficult to evaluate the findings, and we suggest expanding the manuscript to strengthen the evidence for the claims that are made. All reviewers wanted more documentation of injection sites (volumes and cell counts), and more evidence for truly comprehensive mapping. This was seen as critical for supporting the main potential strength of this paper as a comprehensive atlas study.

As suggested, we have now structured the manuscript as a full Research Article adding additional experiments, main and supplementary Figures, and agree that this allows us to more fully elaborate on the many implications that follow from our results.

2) While all three reviewers supported the paper in principle, and agreed that it presents an interesting angle on topography and convergence of cortico-cerebellar connections, they each struggled with the data interpretation in light of limitations of the technique. The fundamental problem was that the experiments are unable to indicate the brainstem origin of the MF projection patterns. They may be mainly from the pons, but it is not possible to say if any of the other brainstem structures labelled are also sources. The reviewers came up with two suggested experiments to address this concern:

We have performed a number of new experiments that provide additional information regarding the brainstem origin of the mossy fiber projections. These are described in detail below, and also further in the specific responses to the three reviewers.

a) In the spirit of an "atlas paper," the authors could confirm pontine projections to the labeled and unlabelled (A1) regions of cerebellum using retrograde AAV (to rule out extrapontine projections). This could validate the method and strengthen novel findings such as pontine projections to lobes IV/V.

We have confirmed the finding of pontine projections to lobules IV/V by using retrograde tracers as suggested (see new Figure 6 and Figure 6—figure supplement 1).

We have further completed retrograde injections in the key cerebellar regions identified as hubs for multimodal spatial convergence (see new Figure 6 and related supplementary Figures). However, It is difficult to know specifically which A1 receiving cerebellar regions (‘unlabelled’ through pons) would not actually receive pontine projections via other cortical areas and hence still result in substantial pontine labelling via retrograde injections (e.g. see lobule VII injections and Figure 6 – supplementary Figure 2). Pontine and extrapontine pathways may have overlapping terminal fields in the cerebellum – hence, success in separating them spatially with retrograde injections would be unlikely. This is also evident from our retrograde injections in Sim, Crus I, and lobules IV/V, VI, and VII where in each case we found retrogradely labelled cells in both pontine and numerous extrapontine nuclei (see new Figure 6 and Figure 6—figure supplement 1 and Figure 6—figure supplement 2).

Furthermore, we do not think that extrapontine projections should be ‘ruled out’ in this regard – but rather that these pathways are actually likely to contribute (in combination with pontine pathways) to disynaptic cerebrocerebellar input (see Discussion section; reviewer 1 point 1). Of course, this makes these pathways more complicated to separate on the individual level, and the feasibility of making anatomical injections in each separate precerebellar nuclei to trace individual mossy fibers is quite low (e.g. due to difficulty in targeting these brainstem nuclei without approaching through the cerebellum, see also point 2b below). However, we feel that the addition of our new experiments as a whole have added substantial information regarding the potential for both pontine and extra-pontine cerebrocerebellar pathways and the implications of this are also discussed in more detail in the revised Discussion section.

b) Another possibility could be to do dual viral injections: (transsynaptic) AAV-cre in cortex, and AAV-floxed-XFP in pons. If feasible, this could confirm the presence of certain cortical-pontine-cerebellar projections.

Due to the technical challenges inherent in this approach as well as some limitations specified below, we opted for the strategy described above (see point 2a). Specifically, to reach the pontine nuclei you must either go through the cerebellum or cortex – creating potential confounds through the chance for AAV leakage of the floxed-XFP construct. Additionally, one would need 100% transfection rate in the pontine nuclei to be able to conclude that any RFP labelled mossy fiber terminals were from extra-pontine sources. If even one or two pontine cells expressed RFP but not the floxed-XFP, then we would still be unable to conclude the specificity of this pathway; 100% transfection rates with AAVs are rarely feasible – given the rostral caudal extent of labelling possible in the pons, even less so. Finally, to accurately determine the likelihood of each separate precerebellar nuclei projecting as intermediate connections between the cerebral cortex and cerebellar cortex we would have to complete this strategy (AAV-floxed-XFP) for all noted individual precerebellar nuclei. This strategy was not feasible – due to the same issues as stated above, as well as the need for targeting each nuclei without spreading to surrounding structures but still resulting in enough labelling to double label mossy fibers.

In short, we fully agree that the precise brainstem origin of these mossy fiber terminals is of great interest. The most feasible strategy (retrograde injections in the cerebellar cortex, see point 2a) allowed us to determine with greater precision the potential for the specific extrapontine precerebellar nuclei that receive corticofugal input to, in turn, provide input to key cerebellar regions of multimodal spatial convergence. We believe that this additional information provided by our study will be vital to guide many future studies specifically targeting individual pathways for more in depth polysynaptic circuit analysis. This has now been discussed throughout the manuscript and many potential avenues for future research pointed out in the Discussion section.

3) It was agreed that the conclusions regarding potential multimodal integration in individual granule cells were not adequately supported by the data. Rather, this was seen as one possible (and reasonable) interpretation based on a general overlap of MF labelling. Although reviewers (particularly #1) suggested specific experiments that could strengthen this claim, we do not think that it is critical to demonstrate this kind of convergence in the current manuscript. The demonstration of broad overlap itself is interesting. However, the conclusions regarding ultimate convergence should be toned down. The current findings can be presented as broadly consistent with the idea of multimodal integration in individual granule cells, which has been previously suggested by earlier studies (eg Huang, Ishikawa…).

We agree that this is a specific point of interest and also should be explored in future studies – particularly with further development of circuit tracing tools combined with electrophysiological techniques. Here, we have been more careful to describe convergence as regional or spatial in nature, rather than at the cellular-integration level, and have expanded our discussion with regard to the individual granule cell level with reference to previously published literature which was already cited (e.g. Huang, Ishikawa, Hantman, etc) and including additional evidence from Markwalter et al., 2019 (subsection “Neuroanatomical injections”).

Reviewer #1:In this manuscript, Henschke and Pakan describe a beautiful set of experiments in which they used an AAV2.1-based anterograde, fluorescence protein-based tracing technique (Zingg et al., 2017) to elucidate cerebro-pontine-cerebellar pathways. They focused on primary sensory (S1, V1 and A1), motor and one association area (PPC). They observed, as in previous studies, that these regions target ipsilateral basal pons in a spatially segregated manner. In turn, cerebellar mossy fiber (MFs) are found to be labelled in specific regions cerebellum including both in the vermis and lateral hemispheres. These results are consistent with other tracer and electrophysiological studies. Interestingly they find that A1 does not provide pontine-cerebellar projections, but dorsal auditory cortex does. This contrasts previous results in other species. Finally, and most novel, the authors examined the spatial overlap of MF projections from different cortical regions and show that in some regions multimodal processing is likely. While this has been shown for single granule cells in vivo in Crus I (Ishikawa et al., 2016), a survey of what cerebellar regions are likely to receive multimodal cerebro-pontine input had not been shown previously. In summary, the authors provide provoking evidence of a strategy to specifically label and manipulate different that pontine-cerebellar circuits will be possible.The technique and results described in this manuscript are illuminating, and the data thoroughly analyzed. The manuscript, therefore, has the potential to serve as a thorough atlas for cortical projections to precerebellar nuclei. This study paves the way for future studies to refine our understanding of how efference copies and other cortical information might be used by cerebellum to perform accurate sensory motor actions. Unfortunately, limitations in the method preclude a clear biological conclusion from the current dataset, beyond the notion that cortical-pontine-cerebellar wiring is similar in mice as other species:1) The authors show that AAV1-cre injections in most cortical regions tested indeed label the pons, but also extra-pontine precerebellar nuclei. This highlights the fundamental problem that we cannot be sure that the MF projection patterns in cerebellum are solely a result of pontine projections -as implied in the mansucript. One example of a new biological conclusion is that cerebro-pontine projections were largely thought to be restricted to posterior and lateral cerebellum, but here the authors seem to show prominent projections in lobes 4 and 5. However, we cannot be sure of this conclusion because the MF labelling in lobes 4 and 5 could be from another precerebellar nucleus. One possible experiment to confirm that pons projects to a given region of cerebellum would be to perform retrograde labelling from different cerebellar regions identified in the anterograde transynaptic method, and verify that cell bodies in pons are labeled.

We performed additional tracing experiments with injections of the retrograde tracer Choleratoxin-B (CTB) into lobes IV/V, VI, VII, Sim and crus I of the cerebellum, as well as using a dual injection approach as a proof of principle to show double labelled cells in the pontine nuclei from injections of the trans-mono-synaptic AAV1.cre into M1 and retrograde CTB injections in lobule IV/V (Figure 6 and Figure 6—figure supplement 1, Figure 6—figure supplement 2). Our results show that the majority of retrogradely labeled cells were located in the pons, with some cells also double labeled, i.e. direct confirmation that these pontine cells receive disynaptic input from the specified cortical regions. We also note, however, that many precerebellar nuclei identified as receiving cortical projections also contained retrograde labelling in these key regions (e.g. lateral reticular nucleus (LRt), reticulotegmental nucleus (RtTg), vestibular nucleus (VN), interpolar part of the spinal trigeminal nucleus (Sp5I) and matrix region x (Mx) (see new Figure 6 and Figure 6—figure supplement 1). Although double labelled cells were not directly observed in extra-pontine nuclei, the labelling was spatially overlapping (Figure 6B) and the proportion of retrogradely labeled cells in extra-pontine nuclei were particularly substantial from lobule IV/V and VI (Figure 6—figure supplement 1). Given the low density of trans-synaptic mossy fiber labelling and the spatially restricted injection sites, we believe the probability of observing double labelled cells within these precerebellar nuclei was very low; however, importantly, these results indicate the potential for specific precerebellar nuclei to either relay or integrate cerebrocerebellar information. Therefore, we conclude that cortical information is likely to reach the cerebellum through both pontine as well as extra-pontine intermediate nuclei. This is now discussed in detail in the revised manuscript (e.g. Discussion section).

2) Another conclusion proposed by the authors is that multimodal integration is likely in certain cerebellar regions. While overlapping domains of MF projections is necessary to achieve multimodal integration. A critical question is whether single granule cells receive different cortical inputs. Such multimodal combinations have been described for motor and proprioception (Huang et al., 2015), for visual, somatosensory and auditory (Ishikawa et al., 2015), as well as visual and vestibular information (Chabrol et al., 2015). Understanding the role of multimodal processing in cerebellum requires knowing whether cortical information is "mixed" at the level of single granule cells, as proposed by Marr and Albus. Are there specific wiring rules for cerebro-ponto-cerebellar projections? For example, perhaps corollary information only pairs with its principal sensory modality partner. Unfortunately, these experiments are more challenging and would require two-color labelling using another conditional method (e.g. Flp), in addition to a strategy to sparsely label GCs (electroporation, patch or mouse lines (Huang et al., 2013)). Nevertheless, this would significantly advance our understanding of how cortical information is processed at the input layer of cerebellum.

We agree that this is a very interesting question, and indeed, difficult to address fully at this time with our current tools. A very recent proof of principle paper has just been published (Zingg et al., 2020) using similar techniques to that suggested, with Flp and cre dual labeling, so this may be a strategy we can also pursue in the future given the availability of AAVs, mouse lines, etc. Regardless, for any future in vivo study of multimodal cellular integration in this regard, targeting in the cerebellum will also be difficult since the density of labelled mossy fiber projections is limited, one would need to maximize the likelihood of targeting investigations to a region (and even a specific module) with high spatial overlap of cerebrocerebellar terminals. This was a large motivation for the current study and is where we see a main strength – as a vital first step/road-map towards future in vivo experiments examining the convergence of cortical input in the cerebellum; making the current manuscript a unique resource moving forward for the field as a whole (see also response to editor, point 3).

3) The lack of cortico-pontine projections into cerebellum from A1 is surprising, in light of previous work. Thus, it is necessary to independently confirm this result using retrograde labelling strategies. If true, a better discussion of why it might be different than previous results is merited. Finally, another concern about the A1 experiments, is that in Figure 1—figure supplement 1 AAV1-cre injections there is very little labeling of inferior colliculus. As I understand the method, retrograde labelling is also expected. As the exact mechanism of transynaptic propagation of the virus or Cre is not known, is it possible that the method failed for this brain region?

To further support our finding of a lack of cortico-pontine projections from A1 we performed two additional sets of experiments. First, we injected a retrograde CAV.cre virus into the pons in the tdTomato reporter mice and found numerous retrogradely labeled cells in the dorsal (AuD) and ventral (AuV) auditory cortex but none in the primary auditory cortex (A1). This finding confirms our previous results that there is no mono-synaptic cerebro-pontine pathway originating in mouse A1.

However, there is a disynaptic pathway connecting A1 with the IC and the pons. This was already evident from our previous experiments, where we found mono-trans-synaptic labeled cells (most likely not retrogradely labeled cells; see below) in the IC. These cells were mainly found in the ECIC and DCIC but rarely in the CIC, as shown in Figure 2 of the revised version of the manuscript, which better illustrates the large amount of labeling in the ECIC/DCIC; therefore, we do not feel as though this method failed along this particular pathway and this is now further demonstrated. Additionally, in order to provide further support for a disynaptic A1-IC-pons pathway, we injected the anterograde trans-monosynaptic AAV1.cre virus into the IC. Following these injections, we found labeled cells in the pons and subsequent MFs in the cerebellum (Figure 1—figure supplement 2).

Based on the results of these two new sets of experiments we conclude that there is indeed not a direct cerebro-pontine pathway to the cerebellum from A1, but instead likely a trisynaptic pathway via the IC. Although the extent of this trisynaptic pathway is yet to be precisely determined, we discuss these findings and outline the potential pathways in detail in the revised manuscript (subsection “Anterograde tracing of indirect cerebrocerebellar pathways”; Discussion section).

Regarding the potential retrograde transport of the AAV1.cre virus, we note that there is no reported projection from IC directly to A1 in mice, this pathway is ‘top-down’ from A1 to IC and not reciprocal, rather information from IC travels to A1 via the thalamus (see also Zingg et al., 2020 where this pathway is chosen specifically for its unidirectional nature). We also note that following our AAV1.cre injections into the IC, we found only a couple labeled cells in the auditory cortex of each experimental animal – very likely representing the extent of retrograde capability of this AAV. This is also in agreement with the original Zingg and colleagues papers (2017; 2020), demonstrating the retrograde transport of this virus is not substantial and far less efficient than in the anterograde direction. These considerations have been clarified in the revised manuscript (see subsection “Neuroanatomical injections”).

4) When examining table 2, I noticed that similar numbers of labelled precerebellar cell bodies can produce differences in the number of MFs by nearly a factor of 10. Does this discrepancy limit the ability normalize for labelling efficiency and thus limit quantification of the results? How could this happen?

We do not wish to downplay the variability in the quantity of labelling we see after viral injections, and hence have mentioned this is in a number of places (including providing the data in the revised Supplementary file 1). However, we do see a significant correlation between the number of anterogradely labeled cells within precerebellar nuclei and the number of mossy fiber terminals (this has been made more clear in the revised manuscript, see subsection “Intermediate cerebrocerebellar brainstem pathways”; Figure 6—figure supplement 3) and a surprising amount of consistency across our cases in the pattern of observed labelling. We have made efforts to present the data in both normalized (scaled [Figure 3] and unscaled [Figure 4]) and raw (Supplementary file 1) format to avoid any misleading conclusions based on normalization.

We believe the variability in absolute numbers is a property of the virus/trans-synaptic transport which is highly dependent on the titre, not only of the injected solution, but subsequently that gets taken up into individual cells; which, in turn, is dependent on a number of factors: volume, spread, speed of injection, etc. For instance, an individual cell that contains a high virus titre will result in transport, however, a neighboring cell that still contains virus at the injection site, but at a low titre, will not contribute to the trans-synaptic quantification. We believe this introduces some inherent variability across cases, even with similar injection site total volume. We have expanded our description of the potential factors to be aware of during injection in this regard (see subsection “Neuroanatomical injections”).

5) The conditions used to perform imaging should be better described. For example, how was high-resolution imaging performed with a 2.5 x objective? What was the NA of this objective? Given the accuracy of the galvanometer mirrors, necessary zooms may not be possible to achieve true diffraction limit resolution. The pixel and image sizes should be mentioned.

We have added the appropriate information into the revised manuscript: Sections of 40µm thickness were examined using a confocal microscope (Zeiss LSM 700, Germany) equipped with a 2.5x objective (NA 0.085, Zeiss, Germany). For each section throughout the extent of the cerebellum two high-resolution images (2048 x 2048 Pixel) were acquired, i.e. one for each hemisphere. The two tiles were merged using custom written MatLab scripts (Mathworks, MA, USA; resulting in a ~4000 x 4000 pixel image of the cerebellum). See subsection “Data analysis”.

6) How was the injection volume measured? The width of the column size measure was presumably estimated from a line profile along an image, but what width parameter was taken as a width. The images shown are clearly saturated. Is this the same image in which the size of the infected region was estimated?

We have added further information regarding the exact parameters used to determine the injection site volume, including a supporting Supplementary file 2A: The injection sites had a roughly cylindrical shape. All sections that covered the central core but not surrounding halo of each injection site were included in the analysis (see Supplementary file 2). We then calculated injection site volume as follows: *V = π*a*b*h*, where *h* corresponds to the maximum value across sections for the height of the injection site [dorso-ventral direction], *a* corresponds to the maximum value across sections for the radius of the injection site parallel to the cortical layers [medio-lateral direction], and *b* corresponds to the rostrocaudal extent of the core injection site. The Line Measure Tool of the Zen software (Zeiss, Germany) was used to measure *h* and *a*, and *b* was calculated by counting the numbers of sections covering the core injection site multiplied by their thickness (40 µm). See subsection “Data analysis”.

Reviewer #2:This is a short report using a trans mono synaptic anterograde tracer method to chart cerebro-cerebellar anatomical projections in mice. Injections into a number of different neocortical regions (M1, S1, V1, A1, posterior parietal association cortex or the dorsal field of auditory cortex) revealed topographically organized projections to the pons and mossy fibre terminals in the cerebellar cortex. The report is generally well written and provides a valuable 'road map' for future studies of the functional significance of cerebro-cerebellar projections in mice.Essential revisions:1) The report needs to make clear what are the novel findings.

The main strength of our work is the description of the multiple disynaptic pathways, in all their complexity, in a single study directly comparable across brain regions. Here we go beyond a focus simply on traditional pontine projections and also detail the various precerebellar nuclei that receive both direct cortical input and project to the cerebellum, for corticocerebellar pathways originating from multiple sensory and motor processing brain regions.

Sorting out all the specific disynaptic and trisynaptic pathways observed in our findings and the functional implications is now perhaps a years-long endeavor – building on the vast amount of solid anatomical work that has come before us, with the ever advancing circuit tracing tools and functional markers now at our disposal. We have confirmed previous hypothesized results, removing ambiguity of terminal fields where functional synaptic connections were unclear, and also provided data from mice that had only previously been investigated in other mammalian species. We strongly believe that this study provides a critical starting point to further investigations and we detail a number of implications from our findings that should be pursued. While this strength of our paper is perhaps not a traditional ‘novel finding’ that can be easily pinpointed by bullet points, we have also uncovered novel findings along the way and have made efforts to highlight these in the revised manuscript, particularly throughout the discussion.

2) How did differences in cell counts and other measures of injection site affect the pattern and extent of anterogradely labelled cell and mossy fibre terminals? It was not clear to me if Figure 2A shows the total extent of injection sites into different neocortical areas or just the defined regions of interest. Either way the area clearly varies considerably between targets which will affect projection density and possibly topography.

We have provided further clarification of the injection site extent (revised Figure 3A; additional Supplementary file 2A), relationship between the injection site volume and mossy fiber terminals (additional Supplementary file 2B) as well as cell counts and mossy fiber terminals (Figure 6—figure supplement 3); including explanation throughout the text (e.g. subsection “Intermediate cerebrocerebellar brainstem pathways”). In short, we found that the number of all anterogradely labelled cells correlated most strongly with the number of mossy fiber terminals and measures of injections site volume were a poor predictor of general extent of mossy fiber labeling.

This may be due to a number of factors related to the fact that the trans-synaptic properties of the virus are heavily reliant on having a high titer concentration and, therefore, a high concentration of virus taken up by individual cells; the number of viral particles within each cell is of course difficult to assess and may be reliant on other properties such as the speed of injection, properties of diffusion within the cortical layers, and proximity to the pipette tip, for example – making the number of cells that are transfected to a level amenable to transsynaptic transport variable, even across injection sites with equal volumes. In fact, one may argue that, given the same volume of injected solution, a smaller injection site would indicate a higher concentration of viral particles and therefore predict a negative correlation between injection site and resulting trans-synaptic labeling.

In reality, there are a number of factors at play that may lead to the variability we see here. Of course, although the absolute number of resulting anterograde and terminal labelling is still of interest, this is one reason why for this study we have opted to inject in a limited number of pre-defined key cortical regions, and to normalize the mossy fiber labelling we have observed across regions to present the data in a way that the pattern of resulting labelling can be appreciated independently of the variability we see across injection sites. We have made considerable effort to make this clear in the revised manuscript with the new Figures noted above and further discussion (e.g. noting the need for further targeted exploration of neighboring somatotopically organized regions, subsection “Diversity of cerebrocerebellar pathways”).

3) The histogram axes in Figure 2D are not all drawn on the same scale making comparisons difficult. Also, percentage values are potentially misleading given the large variation in cell counts and mossy fibre terminals listed in supplementary Figure 1B.

We understand the reasoning of the reviewer, it would be ideal to have the same scaling in all graphs for the revised Figure 3D. However, the percentages of labeled mossy fiber terminals varied substantially across the individual cortical areas and we felt that putting them on the same absolute scale would detract from subtle, but still appreciable, differences observed in the pattern of labelling across various lobules for the less well studied cortical targets (i.e. beyond M1 and S1). To overcome this issue and to directly compare the magnitude of inputs of various modalities into the different lobules we choose the format depicted in the revised Figure 4A,B, as well as including the density measures in Figure 4C (and Supplementary file 1C for reference), which are more scalable reflections of the input given by each cortical target region. In this way, both the pattern of expression across all areas and the biases for cortical contribution to individual cerebellar regions can be appreciated. We have also made a note in the manuscript regarding the interpretation of data from regions with very low-density terminal labelling (subsection “Regional convergence of multimodal cerebrocerebellar input within lobules”).

4) Given that the pons is the primary source of mossy fibre inputs, how did the authors determine that other brainstem nuclei also containing labelled cells in the same experiments necessarily provided mossy fibre projections?

While further investigation is required in relation to individual pathways in this regard, we now provide different lines of evidence showing the potential for specific extra-pontine precerebellar nuclei to act as cerebrocerebellar intermediaries:

- Since no pontine labeled cells were observed after A1 injections, but mossy fiber terminals were still found in the cerebellum, one can conclude that at least for this disynaptic pathway, extra-pontine intermediate nuclei play a large role.

- We observed labelling in the inferior cerebellar peduncle as well as the middle cerebellar peduncle, which is now shown in revised Figure 2.

- Our observations from retrograde tracer injections into key cerebellar lobules show a spatial overlap between trans-synaptically labelled precerebellar cells and retrogradely labelled cells from cerebellar injections.

We have detailed this evidence throughout the Results section and discussed in detail in the revised Discussion section (e.g. subsection “Diversity of cerebrocerebellar pathways”; see also reviewer 1, point 1) and note that further investigations into the role of these precerebellar nuclei as relay versus integration centers for cortical signals will make for interesting future studies.

5) The topographical organization within the pons should be compared to the principles of organization defined in previous studies (eg Leergaard, 2003; Odeh et al., 2005).

We significantly expanded our discussion in this regard in the revised version of the manuscript in both the Results section and the Discussion section.

6) The discussion should consider the extent to which cerebro-cerebellar pathways are conserved across mammalian species. How useful is the mouse as a model for study of cerebro-cerebellar projections?

We have added comparison to other species in various applicable places throughout the discussion (e.g. subsection “Diversity of cerebrocerebellar pathways”) and included a statement regarding the role of the mouse as a model for cerebrocerebellar projections (subsection “Convergence of functionally distinct cerebrocerebellar input”).

Reviewer #3:Henschke and Pakan have submitted a manuscript entitled "Convergence of multimodal inputs connecting the cerebral cortex to the cerebellum" in which they use modern neuronal tracing methods to map circuit projections into the mouse cerebellum. The authors traced the cerebellar inputs originating from sensory, motor, and association cortices. They argue that diverse functional connections project from the cortex to cerebellum, where they ultimately exhibit a convergent terminal field distribution within specific lobules. The manuscript is well written and technically, it offers a much-needed approach to better understand the organization and trajectories of major cortico-cerebellar projections. In addition, the figures are well made, schematics very helpful, and overall the panels are easy to follow. However, below we provide a number of comments and concerns that we hope will improve the clarity and impact of the study.1) Throughout the text, the discussion of A1 and its unique projection through non-pontine pre-cerebellar nuclei is often highlighted, however the significance of this point is not well articulated. Further clarification in the abstract and in the body is needed if the authors wish to highlight the uniqueness of this pathway. This is especially true given the findings in the text that S1 and M1 both have substantial, if not even majority in the case of S1, projections to non-pontine pre-cerebellar nuclei.

We have significantly expanded our discussion in regard to the unique A1 pathways and added additional experiments to support our findings. See subsection “Anterograde tracing of indirect cerebrocerebellar pathways”, subsection “Diversity of cerebrocerebellar pathways” (see also reviewer 1 point 3 and point 4 below).

2) Introduction. Please clarify why the identified cortical regions were selected. In addition, while we agree that the characterization of the injections is rather comprehensive, there is a limitation in this study in that either the number of cortical regions chosen or the breadth of the characterization of the selected regions (ie: why weren't at least two neighboring sites in a given cortical region analyzed) prevent the present study from providing a fully comprehensive map, as stated by the authors.

We agree that the term ‘fully comprehensive map’ was misleading, as the goal of the study was to describe, in detail, only key sensory and motor cerebrocerebellar pathways – consequently, we have toned down these terms throughout. Indeed, two separate strategies could be to (1) briefly characterize many regions without much breadth, or (2) to fully characterize immediately neighboring regions with great detail; however, we opted for a strategy in between, so that we could devote enough space to fully explore the potential for multimodal overlap along with the implications that arise from our findings.

We strongly feel that this type of experimental design is advantageous as it allows us to compare within a single study a number of key functional regions without creating a dataset that acts as only a resource and must be fully interpreted by the reader alone. There is clearly a tradeoff between the number of cortical sites that could be investigated and the detail in which those specific pathways can be highlighted and discussed in depth. We believe that the necessity still exists for more detailed topographic mapping within single modalities, as well as large scale fully comprehensive mapping of pathways (as is provided, for instance, by resources such as the Allen brain Institute), but the strength of our study allows for both the presentation of a large dataset spanning key cortical regions as well as detailed quantification and interpretation of both the results, and identification of key avenues of future research in light of the findings. We believe that with the expansion of the manuscript into a full Research Article format we have achieved this.

3) Figure 1A. Please indicate the circumscribed subregion in each cortical area where the injection was made.

We have further specified the core and halo region of the injection site within the cortical subregions in the revised Figure 3.

4) Subsection “Anterograde tracing of indirect cerebrocerebellar pathways”. Please clarify percentages for A1 projections in order to make the second sentence "This pattern of extra-pontine labeling was similar…" clearer. Further, in Figure 2B, while the scarcity of the projections to the pontine nuclei is similar, they seem to be qualitatively different in that A1 projects terminals, while the AuD experiment labels cell bodies. This distinction is not clear in the text. Moreover, the fact you see terminals in one region and then cell bodies another makes it a challenge to piece together what the actual pathway(s) might look like. That is, how do you substantiate the claim that the cell bodies in the pons (the unexpected labeling) actually project to the cerebellum? And on the flip side, for the majority of S1 labeling, which is in the precerebellar non-pontine nuclei, can we really assume that these all project to cerebellum? If not, what percentage?

The sentence in question has been revised in conjunction with the expanded discussion of the projections from A1 throughout, specifically clarifying the difference between the terminal labelling we see in the pons after A1 injections and the cell bodies observed after AuD injections (with supporting evidence from retrograde labelling from the pontine nuclei [Figure 1—figure supplement 1] and mono-trans-synaptic anterograde labelling in the pons following IC injection [Figure 1—figure supplement 2]). See subsection “Anterograde tracing of indirect cerebrocerebellar pathways” and subsection “Diversity of cerebrocerebellar pathways“ (see also reviewer 1 point 3). We have also further quantified the percentage of extra-pontine versus pontine labeling observed after targeted retrograde injections (new Figure 6—figure supplement 1).

5) The supplementary file 1 is a bit difficult to interpret. Please include the percentages referenced in the text in the table. In addition, it is unclear why in some injection sites the sum of the individual pre-cerebellar nuclei cells equals the summed total, while in other cases the sum is less than or greater than the listed total. (ie M1-1 52 vs 55; V1-1 8 vs 8; A1-1 10 vs 8)

We have included the percentages in the revised Supplementary file 1B (total cell count and mean ± s.e.m. for the percentage of extra-pontine labeled cells for each target cortical region, as discussed in the text). We have also amended the sums, as these were typos based on a previous version of the Table.

6) The significance of terminals in pontine/pre-cerebellar nuclei without cell body labelling is unclear and there should be an explanation of this in the text. For instance, in regard to A1/AuD does this indicate inefficiency in trans-synaptic labelling, that there is an alternative monosynaptic projection from A1/AuD to the pontine nuclei, or is there an alternative explanation? This is mentioned for M1 projections to the olivary nuclei (Subsection “Anterograde tracing of indirect cerebrocerebellar pathways”).

We have performed additional experiments (e.g. IC injection) and expanded the discussion regarding these points in the text (in the subsection “Anterograde tracing of indirect cerebrocerebellar pathways” as well as the subsection “Diversity of cerebrocerebellar pathways”). We indicate the potential for polysynaptic pathways and discuss likely routes for cerebral information from A1 to get to the cerebellar cortex through higher-order polysynaptic pathways. Clearly, deciphering all trisynaptic pathways from all cortical target regions is beyond the scope of the current study, but here we provide not only a starting road-map for these targeted investigations, but also functional ties and implications in the discussion (e.g. in relation to A1/AuD, subsection “Diversity of cerebrocerebellar pathways”; in relation to olivary projections, subsection “Diversity of cerebrocerebellar pathways”).

7) In the subsection "Convergence of multimodal cerebrocerebellar input within lobules" the metric for dominant representation (> or < 55%) is problematic given the relative sparsity of the injections (either across or within cortical regions). Furthermore, given the variability of numbered terminals across individual animals in a given cortical area (ie from 2406 to 25616 from the M1 injections), this type of quantification and conclusions about convergence seem premature.

We agree with the reviewer and have replaced this figure with a more quantitative assessment of density (Figure 4C and Supplementary file 1C). This still demonstrates the point that motor and somatosensory cortical projections in general dominate throughout the cerebellar cortex, without setting arbitrary thresholds or presenting misleading summaries per lobule based on narrow cortical target regions. Additionally, we have included a note in the manuscript regarding the interpretation of data from regions with sparse terminal labelling (subsection “Regional convergence of multimodal cerebrocerebellar input within lobules”).

8) While the discussion about convergence and integration is quite interesting to speculate upon, it is hard to grasp whether the current data is sufficient to reach such a conclusion. Given the vast number of granule cells and the fine scale nature of cerebellar microzones, conclusions about integration of multimodal afferent information by single units (whether microzone or granule cell) cannot be easily drawn. To do this, a method distinct from the approach used in this paper would be needed whereby labelling from two distinct cortical areas could be distinguished in the cerebellum in a single animal. In the end, I am having trouble fully appreciating how integrated the "multimodal" and "convergence" nature of the pathways actually is, based on the data provided.

The concept of ‘convergence’ in this study has been clarified throughout to mean regional spatial overlap and the text has been revised throughout to limit speculation about integration and highlight the potential for modular multimodal processing (see also response to editor point 3). Additionally, we have expanded our discussion regarding convergence on the single cell level with more extensive reference to previously published literature (e.g. see subsection “Convergence of functionally distinct cerebrocerebellar input”).

9) Finally, there is a core assumption at play, which is that cortical injection – pontine/precerebellar cell body labeling – mossy fiber terminal marking. In the case of cortico-ponto-cerebellar pathways, the abundance of literature makes this assumption sound. However, in the case of the precerebellar projections (especially S1 which has a majority non-pontine projections), sagittal sections (for example) demonstrating the axonal projections, and elucidating the monosynaptic pathway would provide a helpful piece of evidence to support the current content.

We thank the reviewer for the suggestion and note that we were able to observe clear labelling in the inferior cerebellar peduncle (even in coronal sections), which has now been included in the revised manuscript (Figure 2; subsection “Anterograde tracing of indirect cerebrocerebellar pathways”). Please also see response to reviewer 2 point 4.

10) In the Abstract, it is not clear what is actually meant by "Prominent terminal overlap across motor and sensory modalities…". And, how was this measured?

We have altered this sentence in the Abstract to read: ‘Within molecularly defined cerebellar modules we found spatial overlap of mossy fiber terminals across motor and sensory modalities…’ and in combination with edits to the last sentence: ‘…the regional convergence of multimodal inputs.’ we have clarified this.

11) It would be very helpful to include the species studied in the Title and/or the Abstract.

The species has now been included in the Abstract.

12) It is critical that the authors fully outline the limitations of the mono-trans synaptic anterograde viral tracer, in their hands, and specifically for the purpose of this study.

This has been made more explicitly clear in the various places in the manuscript (e.g. subsection “Neuroanatomical injections”).

13) In Figure 1B, it is not clear what the red arrows are representing and pointing towards. Aren't the arrows pointing the wrong way if this is to show anterograde tracing direction?

Here, the red ‘arrows’ were indicative of labeled projection neurons (i.e. layer 5 pyramidal cell bodies). These were, however, quite small illustrations, and we have revised this figure for clarity and changed the symbols to circles (Figure 1B).

14) Is there a time dependence on the tracing? That is, do longer periods of survival after injection label additional projections?

Previous studies using this AAV as a mono-trans-synaptic tracer (Zingg et al., 2017) have systematically investigated the time dependence of this technique. They found no significant difference in the number of anterogradely labelled cells between 2, 3, or 4 weeks expression. Indeed, in pilot studies we also confirmed their results by testing different survival times after the injections, up to 4 weeks and, additionally, never observed labelled granule cell/Purkinje cells or other identifiable non-specific labeling – which would indicate transmission beyond mono-synaptically connected neurons. We did not test survival times for shorter than 2 weeks, but this is a fairly standardized time period for any AAV transfection. Hence, 2 weeks expression was used for all the data collected for the main study. We have added a note in the methods to this effect (subsection “Neuroanatomical injections”).

15) In Figure 1F, M1 should project ipsilateral, but in the cerebellum one can see many terminals on both sides. Please explain.

We are unsure as to the specific confusion, but we can clarify that after M1 injections, *pontine* labeling was mostly located on the ipsilateral side, with few cells also labelled on the contralateral side (Figure 1E), as has been previously suggested in rodents examining terminal labelling in the pons (Mihailoff et al., 1985; Leergaard and Bjaalie, 2007). This conclusion is also supported by studies in primates, by using anterograde tracer to confirm additional corticopontine fibers from M1 on the contralateral side with functional synapses (Morecraft et al., 2018). This is now shown here in mice and with trans-synaptic anterograde labelling without confusion regarding the presence of axon terminals or fibers of passage – a significant advantage of our study. This information has been further clarified in the revised text (see subsection “Anterograde tracing of indirect cerebrocerebellar pathways”). However, in the cerebellum, the majority of *mossy fiber* labeling after all injections into the cerebral cortices (including M1) was located on the contralateral site (e.g. M1: ~25% ipsilateral and 75% contralateral; Figure 3C). The prominent contralateral ponto-cerebellar pathways are well documented (e.g. Serapide et al., 2002) but recent studies examining the projections of single pontine cells in rodents have found both ipsilateral and contralateral collaterals, with some even crossing twice (Biswas et al., 2019; this is also documented in the revised manuscript, subsection “Organization of cerebrocerebellar terminals”). The image shown in Figure 1F is representative of the proportion of ipsilateral and contralateral labelling observed in our study (see also Figure 3C,D), as well as in agreement with previous literature.

16) The authors state "…extent M1, indicating the presence of a polysynaptic cerebro-olivary pathway…" What is the literature to support this?

We have expanded on this in subsection “*Diversity of cerebrocerebellar pathways”*.

17) The authors state "We quantified the number of resulting mossy fiber terminals throughout the extent of the cerebellum and found that this was highly variable across cortical regions…" How variable is the size of the infected cell population in the cortex?

It is difficult to determine the precise number of transfected cells within the injection site due to the trans-synaptic nature of the AAV and the density of labeling in the core region. This is why we have opted to report the volume of the injection site core (Supplementary file 1A and Supplementary file 2A). We additionally added a new supplementary file (Supplementary file 2) to illustrate the quantification of the injection site, and further characterize the relationship with the resulting mossy fiber labeling.

18) The authors state "There was a clear and consistent topography of…" It would be helpful to fully define what you mean by topography here. There are many levels of topography, in different brain regions, and this could mean very different things to different researchers.

We have rephrased this for clarity: ‘In agreement with previous studies (Leergaard and Bjaalie, 2007; for review see, Kratochwil et al., 2017), resulting pontine labeling after cortical injections was topographically organized with more rostral cortical regions projecting more medially in the pons and caudal cortical areas projecting towards the lateral extent of the pontine nuclei (Figure 1E, see also Figure 3B).’ (subsection “Anterograde tracing of indirect cerebrocerebellar pathways”).

19) In Figure 3E, what does each data point represent? A single terminal?

For the high density projections, M1 and S1, each dot represents 5-10 mossy fiber terminals, for all other lower density projections each dot represents 1-5 mossy fiber terminals. Because this is a representative schematic to show the modular pattern of labelling, we refer readers to Figure 3 and Figure 4 for quantification of results across cerebellar regions. We have clarified this in the figure legend (revised Figure 5).

20) Along the same lines as above, it would be helpful to see more actual data with the combined zebrin labeling. Mossy fiber patterns, by nature of their clustered appearance, are not so easily appreciated and therefore additional examples, at higher power magnification, would be welcomed.

We revised the current Figure 5 and have added the new Figure 5—figure supplement 1 to the revised manuscript, presenting more AldoC related data. We have also added additional text to the results, specifically detailing the pattern of distributions in relation to the molecular zones (subsection “Regional convergence of multimodal cerebrocerebellar input within lobules”).

21) The authors state "This allowed for a detailed comparison of cerebellar regions across brains and, hence, across functional cortical regions representing multiple modalities (see Figure 3—figure supplement 1)." However, I don't see a full description or explanation of the data in the text.

We have rephrased this to clarify the goal of using the aldolase c expression pattern as a tool to align cerebellar regional maps across target cortical injection sites (subsection “Regional convergence of multimodal cerebrocerebellar input within lobules”). Additionally, as noted above, we have added further description of the pattern of labelling in the text (subsection “Regional convergence of multimodal cerebrocerebellar input within lobules”). Our results were relatively surprising, in that we did not see clear patterns of sagittal organization on a larger scale (which we were anticipating). We believe this is partially due to the fact that we are labelling multiple cerebrocerebellar pathways, and finer scale projections may show more defined sagittal organization. This is now explicitly noted in the revised manuscript (subsection “Regional convergence of multimodal cerebrocerebellar input within lobules”).

22) The number of animals used in the study seems quite low.

We have added a substantial number of additional animals through various new experiments and have almost tripled the number of animals used in this study; we now use 42 animals. Although somewhat subjective for anatomical studies, we would argue that for our experimental design, descriptive statistics, and multiple target regions investigated, the animal numbers we use are in the normal range (for example, Zingg et al., 2017, n = 4 for main anatomical experiments examining a single pathway).

23) In Figure 3—figure supplement 1, what is the red staining in Purkinje cells?

We use the tdTomato Ai9 mice line, which tends to have a level of background fluorescence (see https://www.jax.org/strain/00790, and also e.g. Hahn et al., 2019), as reported by the donating investigator to the Jackson Laboratory: “Importantly, the donating investigator reports that very low levels of tdTomato expression may be present prior to introduction of Cre recombinase – but the tdTomato expression levels after Cre recombination are significantly greater than those baseline levels”.

In our hands, this background staining is very clearly distinguishable, particularly under the microscope, and qualitatively distinct from the AAV1.cre-induced tdTomato expression in cells and terminals. We have added an explanation to this effect in the revised manuscript (Materials and methods section, and revised Figure 5—figure supplement 1 legend).

[Editors’ note: what follows is the authors’ response to the second round of review.]

Essential revisions:The major remaining concern – shared by both reviewers – is the overemphasis/ speculation on 'multimodality' for this essentially anatomical study. Specifically:1) I was delighted to see a more detailed examination of the location of MFs from different modalities within each lobule, since a prerequisite for multimodal processing is that the information project to the same subregion of a lobule. I also found rather striking the case in Crus II where there was still segregation between two MF types within the same lobe. This latter result highlights the notion that multimodal innervation within lobes is necessary but not sufficient to conclude multimodal processing. I therefore encourage the authors to soften their choice of words of "highly multimodal" and "moderately multimodal" as a conclusion of Figure 4. Perhaps the use of the term "lobule co-innervation" is more appropriate. Then after Figure 5, the argument for putative multimodal processing is much stronger, but still only suggestive.

In reference to Figure 4, we have changed the terms here as suggested to utilize ‘coinnervated’ and we thank the reviewer for this suggestion (e.g. subsection “Lobule co-innervation of cerebrocerebellar inputs from distinct cortical areas”, Figure 4).

We have additionally further reworked our concept of multimodal ‘integration’ for this more anatomically relevant term throughout the manuscript where appropriate. Therefore, we have toned down the functional ‘multimodal’ emphasis and highlighted the strength of the work as an anatomical study (e.g. Title; Introduction).

2) Indeed, Figure 5 shows an essential set of analyses suggestive of multimodal processing, but I was left wanting for some clear quantification of "intermingled" and a simple overall summary.a) Would it be possible to analyze the distribution of nearest-neighbor distances between terminals of different origins? Those that co-innervate would have shorter distances and those that do not, e.g. Crus II, would show a shifted distribution. Perhaps a more sophisticated point-process analysis, such as the application of the bivariate Ripley K-function, could be used to test for spatial randomness between two species of points and provide statistical confidence.b) To add some statistical rigor to the alignment with adolase C staining, one could compare the frequency of AldoC positive and negative terminals for each cortical injection. Statistical tests for the frequency of events can then be used.

a) As suggested, we have analyzed the distribution of nearest neighbor distances (Figure 5B and Figure 6B) as well as the identity of the cortical origin of nearest neighbors (Figure 5C and Figure 6C) and included appropriate statistical tests for significance (see also exact pvalues in Supplementary file 1E). Consequently, we have split the previous Figure 5 into two figures due to space constraints (now Figure 5 and Figure 6). The results have been amended to reflect these changes and incorporate the main points from the quantification (e.g. subsection “Spatial organization of mossy fiber terminals in molecularly defined cerebellar modules”). We have additionally included analysis using the Ripley K function to test for spatial randomness and included the results in the text (subsection “Spatial organization of mossy fiber terminals in molecularly defined cerebellar modules”). Of course, the methods section has also been updated to reflect the new analysis (subsection “Quantification and data analysis”). We believe these additional analyses have indeed increased the statistical confidence in the qualitative pattern of data previously reported and thank the reviewer for their detailed suggestions.

b) We have added statistical rigor and internal validation of the AldoC+/- alignment across animals by correlating the frequency of mossy fiber terminals across + and – regions of each lobule. See new Figure 5—figure supplement 1J where we present the correlation matrix across all animals as well as the associated p-values; demonstrating the high correlation across animals with injections within the same cortical target region (referenced in the figure legend as well as subsection “Spatial organization of mossy fiber terminals in molecularly defined cerebellar modules” and subsection “Quantification and data analysis”).

3) The title continues to hint at a more functional set of analyses, which the paper really does not provide. Please revise.

We have revised the title accordingly: ‘Disynaptic cerebrocerebellar pathways originating from multiple functionally distinct cortical areas’. We think this title preserves the concept of functional specificity of cortical inputs, but that it is now also clear that the study is based on an anatomical premise (see also point 1).

4) Similarly, the Abstract could present the key anatomical distinctions that were uncovered rather than the speculative discussion about multimodal sensory input – this functional idea should remain, but it should be toned down to make space to better clarify the actual anatomical findings.

We have rewritten the Abstract to downplay the multimodal integration further and to highlight the anatomical strengths. We agree that it is now much more focused.

5) The main Figure 6 merits a more thorough quantification of retrograde labeling.

We have added further quantification to (now) Figure 7, including a quantitative assessment of the origin of nearest neighbors for anterogradely labelled cells (Figure 7C) as well as the distance between anterograde and retrograde labelled cells from dual injections (Figure 7—figure supplement 2; subsection “Intermediate cerebrocerebellar brainstem pathways”). We also refer to supplementary file 1B which includes a table of the quantification of retrograde labelling and Figure 7—figure supplement 1 and Figure 7—figure supplement 3 as qualitative as well as quantitative representation of this data.

6) Quantification and presentation of co-labeled and spatially intermingled neurons following dual viral tracing injections would be appropriate.

We have added this information in conjunction to the points listed above (point 5), including further quantification and description regarding co-labeled and spatially intermingled neurons (see also Figure 7 figure legend).

7) I like the estimation of the number of MFs per pontine neuron and would like to see it as a main figure.

We have including this into the updated main Figure 7 (see Figure 7D).

8) The Discussion section rambles on a bit. I believe it can be shortened and used to more clearly summarize the major findings, the similarities and differences with previous results, and the implications for cerebellar function.

We have edited the Discussion section throughout to be more concise and clearer (reducing the length by ~500 words), highlighting the major findings while still maintaining a thorough discussion of the nuances of this extensive dataset.

9) The section titles are rather vague. For example, "Organization of cerebrocerebellar mossy fiber terminals" is essentially duplicated in the Results and Discussion, and perhaps needs more specific wording. "Spatial organization of cerebrocerebellar mossy fiber types". I am not really sure why there is a Discussion subsection on "Diversity of cerebrocerebellar pathways" as you define the pathways by viral injection. Perhaps something like: "Cerebellar cortical target diversity of cerebrocerebellar pathways?"

We have amended and refined most section titles. In the Results section, we now move from the more general trans-synaptic anterograde tracing technique and general pathway results, to the regional organization of terminals in the cerebellum, to the co-innervation of inputs at the level of the lobule, to the precise spatial organization of inputs on the level of molecularly defined modules within each lobule, and finally to the summary examining in detail the entire cerebrocerebellar pathway including the intermediate brainstem nuclei.

Likewise, the Discussion section titles have been amended to explore the stages of the cerebrocerebellar pathways, first cortex to precerebellar nuclei (subsection “Pontine and extra-pontine cerebrocerebellar pathways”), then the terminal organization in the cerebellum (subsection “Organization of cerebrocerebellar input to the cerebellum”), and finally the functional implications of the study (subsection “Functional implications”). We believe the general flow is now improved.